# Bilevel Reinforcement Learning for Stock Data with A Conservative TD Ensemble

## Abstract

Reinforcement learning (RL) has shown significant promise in stock trading. A typical solution involves optimizing cumulative returns using historical offline data. However, it may produce less generalizable policies that merely "memorize" optimal buying and selling actions from the offline data while neglecting the non-stationary nature of the financial market. We frame stock trading as a specific type of offline RL problem. Our method, MetaTrader, presents two key contributions. First, it introduces a novel bilevel actor-critic method that spans both the original stock data and its transformations. The fundamental idea is that an effective policy should be generalizable across out-of-distribution data. Second, we propose a novel variant of conservative TD learning, utilizing an ensemble-based TD target to mitigate value overestimation, particularly in scenarios with limited offline data. Our empirical findings across two publicly available datasets demonstrate the superior performance of MetaTrader over existing methods, including both RL-based approaches and stock prediction models.

## 1 Introduction

Reinforcement learning (RL) has demonstrated promising results in stock trading (Deng et al., 2016; Jeong & Kim, 2019; Ye et al., 2020; Briola et al., 2021; Liu et al., 2021; Kumar, 2023; Gao et al., 2023a). Typical approaches initially leverage advanced deep learning techniques to extract useful features from the noisy market data, *e.g.*, stock prices, trading volumes, and financial news. Subsequently, these features are used as inputs for RL algorithms, commonly designed to maximize the expected total payoff within the offline training data. The recent advances of RL-based trading methods, such as StockFormer (Gao et al., 2023a), have shown superior performance compared to straightforward combinations of stock prediction approaches (Li et al., 2018; Xu & Cohen, 2018; Wang et al., 2021; Zheng et al., 2023) with a fixed trading policy, like buying stocks with the highest predicted future gains and holding them for a specific period.

However, a questionable part of most existing methods lies in their direct use of standard RL within offline datasets (see Figure 1), while neglecting the performance of the learned policy in out-of-distribution (OOD) scenarios. As the RL agent cannot further explore the real-world, rapidly changing financial market, it is prone to overfitting the historical data and memorizing the "optimal" offline policy— transactions yielding the highest profits— although it may be impractical beyond the scope of the dataset. This raises a crucial yet under-explored question: *How can we learn more robust trading policies that can jointly handle the in-domain profits[1] and out-of-domain generalizability?*

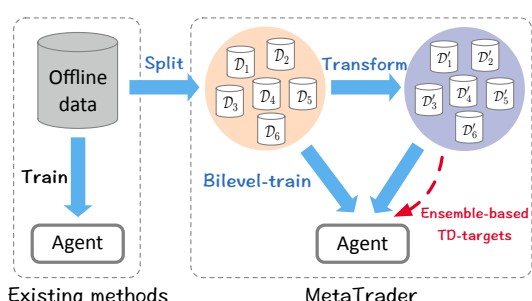

Figure 1: A comparison of MetaTrader and existing RL-based stock trading methods.

In this paper, we propose MetaTrader, an early study of bilevel optimization of actor-critic methods in stock trading, which we formulate as a decoupled offline RL problem. The core idea of MetaTrader

---

[1]In-domain profit refers to the financial gains achieved by the trading policy when applied to data that shares the same distribution as the training data (*i.e.*, the historical dataset).

extends beyond maximizing the expected total payoff on current trajectories. As illustrated in Figure 1, the primary objective includes learning policies that are also effective for OOD stock data.

To achieve this, we improve existing RL traders in two key aspects. One primary contribution of our work is to enhance the generalization capability of the policy. In practice, we approach this from both the algorithmic and data perspectives, which are closely intertwined. From the algorithmic perspective, we propose a new actor-critic method based on bilevel optimization, which has been shown to effectively bridge the distribution shift between the training and testing domains, facilitating the generalization of optimized variables to unseen scenarios (Finn et al., 2017). The intuition of incorporating bilevel gradient updates is to mitigate overfitting to specific historical distribution by explicitly simulating test samples with OOD market conditions, keeping the model from simply memorizing the optimal policy based on specific patterns in the training set.

From a data perspective, we propose specific data transformation methods that aim to generate the simulated OOD test samples for bilevel gradient updates. The data transformation methods are fundamentally designed from different factorized components of the time series data, including short-term randomness, long-term trends, and multi-scale correlations.

Another contribution of our work is a novel temporal difference (TD) method based on an ensemble of transformations of future stock data. This approach aims to make the policies learned from offline data more conservative to address the value overestimation problem. Specifically, we independently compute Q-values separately using both the original data and its transformations at future time steps. We use the minimum Q-value among them as the TD target to supervise value estimation at the current time step. Unlike previous ensemble-based Q-learning methods, which use the multiple target Q-networks to compute ensemble value regularization, our method relies on a single target Q-network and derives the worst-case Q-value through a diverse set of transformed data. We further illustrate the feasibility of this approach from the perspective of Bellman Equations within our offline RL setup.

Our approach significantly outperforms existing RL-for-finance methods on two stock datasets in both portfolio returns and Sharpe ratios, showcasing its ability to balance the trading profits and risks. In summary, this paper presents three contributions:

- We reconsider the rationale behind existing RL-based stock trading approaches, highlighting the risks of in-domain policy overfitting and the problem of value overestimation.
- We leverage bilevel optimization to enhance the generalizability of offline RL across various data transformations, thereby enabling adaptation to the non-stationary environment.
- We introduce a novel ensemble-based conservative TD target to overcome value overestimation.

## 2 PROBLEM DEFINITION

We cast stock trading as a particular offline RL problem. The corresponding Markov decision process (MDP) can be formulated as an 8-tuple $(\mathcal{O}, \mathcal{H}, \mathcal{Z}, \mathcal{A}, P_h, P_z, R, \gamma)$:

**Observation space ($\mathcal{O}$).** The raw data includes: (i) $o_t^{\text{price}} \in \mathbb{R}^{T \times |S| \times 5}$: Daily *open, close, high, low* stock prices, and trading *volumes* for the previous $T$ days. $|S|$ is the total number of stocks. (ii) $o_t^{\text{stat}} \in \mathbb{R}^{|S| \times K}$: $K$ technical indicators that reflect the temporal trends of stock prices. (iii) A matrix that measures the correlations between historical daily closing prices of all stocks.

**State space ($\mathcal{H}$, $\mathcal{Z}$).** Motivated by the observation that individual buying and selling actions typically have limited impacts on market dynamics, we explicitly decouple the state space into two components: $\mathcal{S} = (\mathcal{H}, \mathcal{Z})$. $\mathcal{H}$ is the *action-free* state space that represents the market data, while $\mathcal{Z}$ is the *action-dependent* state space that represents our balance sheet. Accordingly, we formulate the state transition probabilities as $P_h(h_{t+1}|h_t)$ and $P_z(z_{t+1}|z_t, a_t)$. The action-free *market state* $h_t$ is composed of three types of latent states $h_t^{\text{relat}}, h_t^{\text{long}}, h_t^{\text{short}}$ generated from the observation $o_t^{\text{price}}$, $o_t^{\text{stat}}$ and $o_t^{\text{cov}}$ by predictive coding. Please refer to Eq. 1 for details. The action-dependent *balance state* $z_t \in \mathbb{R}^{|S|+1}$ represents the total account balance and holding amount of each trading asset.

**Action space ($\mathcal{A}$).** We use a continuous action space $a_t \in \mathbb{R}^{|S|}$, where each component represents the number of shares to buy, hold, or sell for each asset. To simulate real-world trading, we discretize $a_t$ into several intervals, such as $100, 200, \ldots$ shares when deploying the agent for testing.

**Reward function ($R$).** The immediate reward is defined as the daily portfolio return ratios: $r_t = R(h_{t:t+1}, z_{t:t+1})$, where $z_{t+1}$ is dependent on $a_t$. $\gamma$ is the reward discount factor that determines how much the RL agents care about rewards in the distant future.

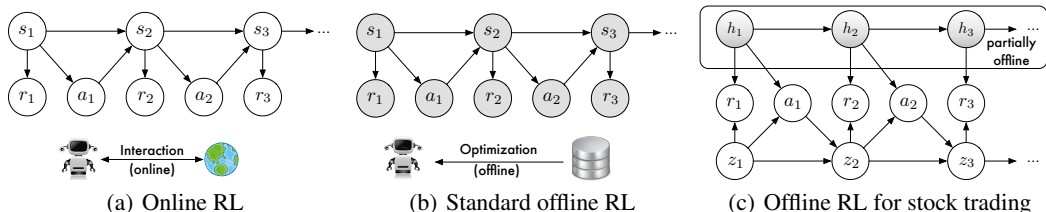

(a) Online RL        (b) Standard offline RL        (c) Offline RL for stock trading

Figure 2: A comparison of different problem setups of RL-for-finance methods. Unlike previous literature, we introduce a novel offline learning setup tailored for non-stationary market data.

**A mixed offline RL setup.** Unlike standard offline RL setups, as shown in Figure 2, we specify the state space as *partially offline*. This means that (i) we can only access limited market dynamics with *unknown* $P_h$, (ii) we assume known action-dependent dynamics denoted by $P_z$, and (iii) we can explore different actions to collect reward feedback with a pre-defined reward function. Offline RL commonly faces challenges related to bootstrapping from out-of-distribution states, often leading to overly optimistic value function estimates. In contrast, our decoupled MDP allows for the online expansion of action-free states $h_t$ using carefully designed data transformation methods. Furthermore, our formulation facilitates an *ensemble-based* TD method for conservative value estimation.

## 3 METHOD

### 3.1 REVISITING RL-BASED STOCK TRADING

We use the recent work of StockFormer (Gao et al., 2023a) as an example. Despite its state-of-the-art performance, a potential drawback lies in the straightforward use of conventional RL methods for offline data. StockFormer has three network branches $f_{\psi_{1,2,3}}(\cdot)$ to extract the cross-stock relational features $h_t^{\text{relat}} \in \mathbb{R}^{|S| \times D}$, the long-term predictive features $h_t^{\text{long}} \in \mathbb{R}^{|S| \times D}$, and the short-term predictive features $h_t^{\text{short}} \in \mathbb{R}^{|S| \times D}$ from the stock data $o_t = [o_{t-H+1:t}^{\text{price}}, o_{t-H+1:t}^{\text{stat}}, o_{t-H+1:t}^{\text{cov}}]$ in the past $H$ days. D represents the dimension of the hidden features per stock. The feature extraction module is frozen during policy optimization. These features are used as the input states of the Soft Actor-Critic (SAC) algorithm (Haarnoja et al., 2018):

$$\text{States: } h_t^{\text{relat}} = f_{\psi_1}(o_t), \quad h_t^{\text{long}} = f_{\psi_2}(o_t), \quad h_t^{\text{short}} = f_{\psi_3}(o_t),$$
$$\text{Actor: } a_t \sim \pi_\theta(h_t^{\text{relat}}, h_t^{\text{long}}, h_t^{\text{short}}, z_t), \quad \text{Critic: } q_t \sim Q_\phi(h_t^{\text{relat}}, h_t^{\text{long}}, h_t^{\text{short}}, z_t, a_t),$$

$$(1)$$

where $z_t \in \mathbb{R}^{|S| \times 1}$ represents the holding amount of all trading assets at a certain time step. Our approach follows the basic network architectures of StockFormer, including the feature extraction module $f_{\psi_{1,2,3}}$, the actor module $\pi_\theta$, and the critic module $Q_\phi$.

A notable concern in previous RL-based stock trading methods, such as StockFormer, is that the RL agent is trained exclusively on maximizing the total payoff within a specific in-domain offline dataset. This approach carries the risk of overfitting the optimal trading behaviors in a fixed dataset, potentially resulting in impractical policies for the unobserved dynamics of a non-stationary market in the future. In summary, there are two primary challenges when deploying RL agents trained on offline datasets to non-stationary financial markets: ***Challenge 1***: Enhancing the performance of the policy in OOD scenarios. ***Challenge 2***: Addressing the value overestimation issues commonly present in offline RL. In the subsequent Section 3.2 and Section 3.3, we delve into the technical details of MetaTrader, offering solutions to these challenges, respectively.

### 3.2 BILEVEL REINFORCEMENT LEARNING ACROSS TRANSFORMED DATA

To improve the generalization ability of the learned policy to scenarios of non-stationary financial markets, we propose a bilevel RL paradigm (see Figure 3), which concurrently considers in-domain rewards and potential profits for OOD data. To simulate the OOD scenarios, we first split the offline dataset chronologically into subsets, and then generate fictitious data with carefully-crafted data transformation techniques. The entire training scheme involves two phases: (i) *OOD policy learning* and (ii) *in-distribution model finetuning*, as illustrated in Alg. 1 and Alg. 2 respectively.

**Subsets construction by data slicing.** Initially, considering the explicit temporal and cyclical patterns present in raw stock data, we partition the entire offline training set into subsets referred to

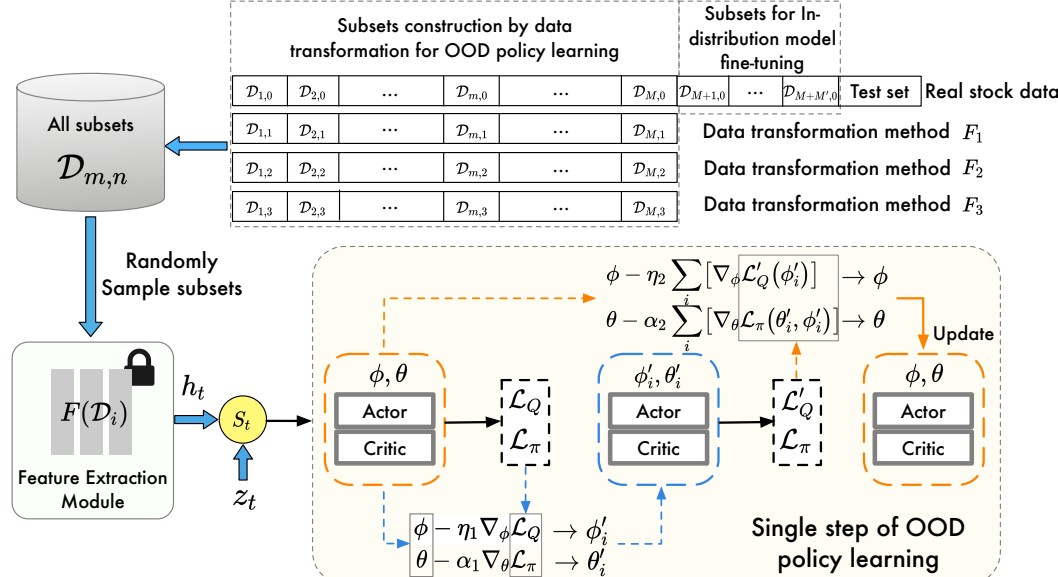

Figure 3: The bilevel learning scheme of MetaTrader based on transformed data.

as $\{\mathcal{D}_m\}_{m=1}^{M+M'}$. As shown in Figure 3, $M$ represents the number of subsets used in the process of *OOD policy learning*, and $M'$ corresponds to *in-distribution model finetuning*. These subsets are then separated into sequences of $64$ days, approximating the number of trading days in a quarter of a year.

**Subsets construction by data transformation.** We believe that the use of stock transformation techniques to expand the offline training data is pivotal for enhancing the model's generalization capabilities. Specifically, we treat stock market data as multivariate time series, those dynamic patterns can be typically viewed as a combination of three components: short-term randomness, long-term trends, and multi-scale seasonal patterns. Accordingly, we introduce three data transformation methods $\{F_n\}_{n=1}^{N=3}$ to simulate OOD yet plausible market changes that have not been included in the training set, with each method focusing on one of the three dynamic components:

- $F_1$: At each time step, we select the Top-$10\%$ stocks (*yellow bars* in Figure 4) with the highest daily gains in prices and invert the growth rate to declines (*blue bars*). It simulates the effects of unexpected events on individual stocks (*i.e.*, short-term randomness). Based on this, our bilevel learning scheme mitigates overfitting to stocks that perform well only within training periods.

- $F_2$: We reverse the overall trends of the stock price. By simulating varying market conditions influenced by long-term disruptions, it evaluates the policy's robustness in such scenarios.

- $F_3$: We downsample the original time series by four. It scales the seasonal patterns of the market, enabling the model to capture multi-scale correlations between the stock changes.

By applying $\{F_n\}_{n=1}^{N}$, we expand the subset collections for OOD policy learning to $\{\mathcal{D}_{m,n}\}_{m=1,n=0}^{M,N}$, where we use $n=0$ to denote the original data. We provide more details in Appendix A.

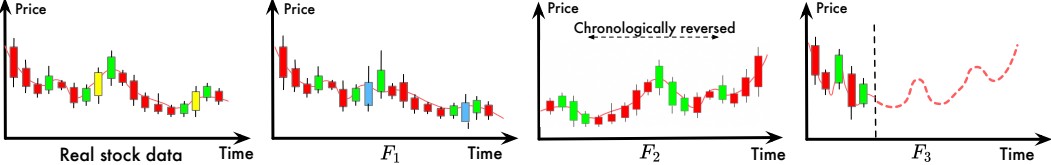

Figure 4: An example of stock data transformation. See text for details.

**Out-of-distribution policy learning.** Based on different partitions/transformations of the original data, we perform cross-set policy learning using a bilevel optimization scheme, as shown in Alg. 1. We first sample training subsets randomly: $\{\mathcal{D}_i\}_{i=1}^{K} \sim \{\mathcal{D}_{m,n}\}_{m=1,n=0}^{M,N}$. The goal of the inner-loop optimization step is to derive a hypothetical RL gradient aimed at maximizing the in-distribution profits within each subset. In the outer-loop optimization step, our model diverges from previous

meta-learning approaches by conducting bilevel gradient updates across each pair of distinct subsets. In *Lines* 10–12 in Alg. 1, we evaluate the hypothetical model parameters $\{\phi'_{k,j}, \theta'_j\}$ that are learned from subset $j$ on the data split $\mathcal{B}_i$ from subset $i$. The goal is to update model parameters to enhance the robustness of the policy to OOD data. In line with SAC, we incorporate double Q-networks $Q_{\phi_{1,2}}$ as well as the corresponding target networks $Q_{\bar{\phi}_{1,2}}$ with moving-average parameters. We will delve into the inner-loop critic loss $\mathcal{L}_Q$, the outer-loop critic loss $\mathcal{L}'_Q$, and the actor loss $\mathcal{L}_\pi$ in Section 3.3.

**In-distribution model finetuning.** Due to the non-stationary nature of the time-evolving market, finetuning MetaTrader on recent training data close to the test set can enhance its final performance. In Alg. 2, we continue to employ the bilevel optimization scheme *within* each training subset. We first draw subsets from the buffer of raw data, such that $\{\mathcal{D}_i\}_{i=1}^K \sim \{\mathcal{D}_{m,n=0}\}_{m=M+1}^{M+M'}$. Notably, we exclusively use the original data to eliminate the unexpected noise introduced by the transformed data. Furthermore, it is essential to note that during the finetuning phase, we perform the inner-loop and outer-loop gradient steps on separate data batches, $\mathcal{B}_i^{\mathrm{tr}}$ and $\mathcal{B}_i^{\mathrm{ts}}$, sampled from the same subset $\mathcal{D}_i$. The rationale behind this is to facilitate the adaptation of the model to nearby market dynamics.

---

**Algorithm 1** OOD Policy Learning

**Input:** Expanded datasets $\{\mathcal{D}_{m,n}\}_{m=1,n=0}^{M,N}$
**Parameters:** $\alpha_1, \alpha_2, \eta_1, \eta_2$
1: Randomly initialize model parameters $\theta, \phi_1, \phi_2$
2: **for** $T_1$ steps **do**
3:     Sample datasets $\{\mathcal{D}_i\}_{i=1}^K \sim \{\mathcal{D}_{m,n}\}$
4:     **for** each $\mathcal{D}_i \in \{\mathcal{D}_i\}_{i=1}^K$ **do**
5:         Sample a batch of data $\mathcal{B}_i \sim \mathcal{D}_i$
6:         $\phi'_{1,i} \leftarrow \phi_1 - \eta_1 \nabla_{\phi_1} \mathcal{L}_Q\left(\mathcal{B}_i;\ \phi_1\right)$
7:         $\phi'_{2,i} \leftarrow \phi_2 - \eta_1 \nabla_{\phi_2} \mathcal{L}_Q\left(\mathcal{B}_i;\ \phi_2\right)$
8:         $\theta'_i \leftarrow \theta - \alpha_1 \nabla_\theta \mathcal{L}_\pi\left(\mathcal{B}_i;\ \theta, \phi'_{1,i}\right)$
9:     **end for**
10:    $\phi_1 \leftarrow \phi_1 - \eta_2 \sum_i \sum_j \left[\nabla_{\phi_1} \mathcal{L}'_Q\left(\mathcal{B}_i;\ \phi'_{1,j}\right)\right]$
11:    $\phi_2 \leftarrow \phi_2 - \eta_2 \sum_i \sum_j \left[\nabla_{\phi_2} \mathcal{L}'_Q\left(\mathcal{B}_i;\ \phi'_{2,j}\right)\right]$
12:    $\theta \leftarrow \theta - \alpha_2 \sum_i \sum_j \left[\nabla_\theta \mathcal{L}_\pi\left(\mathcal{B}_i;\ \theta'_j, \phi'_{1,j}\right)\right]$
13: **end for**

**Algorithm 2** In-Distribution Model Finetuning

**Input:** Datasets $\{\mathcal{D}_{m,n=0}\}_{m=M+1}^{M'}$ of real stock data
**Parameters:** $\alpha_1, \alpha_2, \eta_1, \eta_2$
1: Obtain the pretrained $\theta, \phi_1, \phi_2$ from Alg. 1
2: **for** $T_2$ steps **do**
3:     Sample datasets $\{\mathcal{D}_i\}_{i=1}^K \sim \{\mathcal{D}_{m,0}\}_{m=M+1}^{M'}$
4:     **for** each $\mathcal{D}_i \in \{\mathcal{D}_i\}_{i=1}^K$ **do**
5:         Sample disjoint data batches $\mathcal{B}_i^{\mathrm{tr}}, \mathcal{B}_i^{\mathrm{ts}} \sim \mathcal{D}_i$
6:         $\phi'_{1,i} \leftarrow \phi_1 - \eta_1 \nabla_{\phi_1} \mathcal{L}_Q\left(\mathcal{B}_i^{\mathrm{tr}};\ \phi_1\right)$
7:         $\phi'_{2,i} \leftarrow \phi_2 - \eta_1 \nabla_{\phi_2} \mathcal{L}_Q\left(\mathcal{B}_i^{\mathrm{tr}};\ \phi_2\right)$
8:         $\theta'_i \leftarrow \theta - \alpha_1 \nabla_\theta \mathcal{L}_\pi\left(\mathcal{B}_i^{\mathrm{tr}};\ \theta, \phi'_{1,i}\right)$
9:     **end for**
10:    $\phi_1 \leftarrow \phi_1 - \eta_2 \sum_i \left[\nabla_{\phi_1} \mathcal{L}_Q\left(\mathcal{B}_i^{\mathrm{ts}};\ \phi'_{1,i}\right)\right]$
11:    $\phi_2 \leftarrow \phi_2 - \eta_2 \sum_i \left[\nabla_{\phi_2} \mathcal{L}_Q\left(\mathcal{B}_i^{\mathrm{ts}};\ \phi'_{2,i}\right)\right]$
12:    $\theta \leftarrow \theta - \alpha_2 \sum_i \left[\nabla_\theta \mathcal{L}_\pi\left(\mathcal{B}_i^{\mathrm{ts}};\ \theta'_i, \phi'_{1,i}\right)\right]$
13: **end for**

---

### 3.3 ENSEMBLE-BASED CONSERVATIVE TD LEARNING

Within the aforementioned bilevel learning framework, we formulate the actor's objective function $\mathcal{L}_\pi$ as $\min_\theta \mathbb{E}_{s_t}\left[D_{\mathrm{KL}}(\pi_\theta(\hat{a}_t|s_t) \| \exp(Q_{\phi_1}(s_t, \hat{a}_t))/Z_{\phi_1}(s_t))\right]$, where $Z_{\phi_1}$ is a normalization factor. For the critic loss, we introduce a novel TD method to mitigate the value overestimation issue inherent in offline RL. In Alg. 1, the training objectives of $Q_{\phi_{1,2}}$, including the inner-loop $\mathcal{L}_Q$ and the outer-loop $\mathcal{L}'_Q$, can be formulated as $\min_{\phi_k} \mathbb{E}_{(s_t, a_t)}\left[1/2(Q_{\phi_k}(s_t, a_t) - \widehat{Q}(s_t, a_t))^2\right]$, where $Q_{\phi_k}(\cdot)$ represents the TD estimate of the critic $k$ at timestamp $t$, and $\widehat{Q}(\cdot)$ represents the corresponding TD target. We here denote $s_t = [h_t, z_t]$ and $h_t = [h_t^{\mathrm{relat}}, h_t^{\mathrm{long}}, h_t^{\mathrm{short}}]$ (see Eq. 1). In the inner-loop gradient step, we formulate the TD target as

$$\widehat{Q}(s_t, a_t) = r_t + \gamma\left[-\lambda \log \pi_\theta(\hat{a}_{t+1} \mid s_{t+1}) + \min_{k=1,2}(Q_{\bar{\phi}_k}(s_{t+1}, \hat{a}_{t+1}))\right], \tag{2}$$

where $\hat{a}_{t+1}$ is generated by $\pi_\theta(\cdot \mid s_{t+1})$, $r_t$ is computed based on $\{h_{t:t+1}, z_t, a_t\}$, and $Q_{\bar{\phi}_k}$ is the next-step Q-value from each target Q-network. In the outer-loop gradient step in Alg. 1, we further incorporate an ensemble of TD targets in $\mathcal{L}'_Q$ derived from the transformed data:

$$\begin{aligned}
\widehat{Q}'(s_t, a_t) = r_t + \gamma\big[&-\lambda \log \pi_\theta(\hat{a}_{t+1} \mid s_{t+1}) \\
&+ \min_{k=1,2}\ \min_{n=1:N}\left(Q_{\bar{\phi}_k}(s_{t+1}, \hat{a}_{t+1}), Q_{\bar{\phi}_k}(s_{t+1}^{(n)}, \hat{a}_{t+1}^{(n)})\right)\big],
\end{aligned} \tag{3}$$

where $s_{t+1}^{(n)}$ is obtained from the transformed data by $\{F_n\}_{n=1}^N$ and $\hat{a}_{t+1}^{(n)}$ is generated by $\pi_\theta(\cdot \mid s_{t+1}^{(n)})$.

While the mathematical expressions of our method and other existing methods appear similar, significant differences exist in the network model and data input. Notably, existing ensemble-based

Q-learning methods (An et al., 2021; Lee et al., 2022; Wu et al., 2022) typically utilize multiple target Q-networks (with separate model parameters) and compute ensemble value regularization by exploiting the implicit diversity among these Q-networks. In contrast, our approach relies on a single target Q-network and derives the worst-case Q-value by leveraging the explicit diversity introduced through transformed data. Please refer to the Appendix G for more details.

We here demonstrate the feasibility of the ensemble-based TD method from the aspect of Bellman Equations. In SAC, the soft Q-value is computed iteratively, starting from any Q-function and repeatedly applying a Bellman backup operator $\mathcal{U}^\pi$ given by $\mathcal{U}^\pi Q(s_t, a_t) = r_t + \gamma \, \mathbb{E}_{s_{t+1} \sim P}[V^\pi(s_{t+1})]$, where $V^\pi(s_t) = \mathbb{E}_{a_t \sim \pi}[Q(s_t, a_t) - \log \pi(a_t \mid s_t)]$ is the soft state value function for policy $\pi$. In our offline RL setup, the state transitions can be decoupled as $P_h(h_{t+1}|h_t)$ with *stochastic*, *unknown* action-free transitions and $P_z(z_{t+1}|z_t, a_t)$ with *deterministic*, *known* dynamics. For a transformed data trajectory starting from an original data point, $\tau = (h_t, z_t, a_t, \tilde{h}_{t+1}, z_{t+1}, \tilde{a}_{t+1}, \ldots)$, where $h_t$ is encoded from the original offline data $o_t$, $\tilde{h}_{t+1}$ is encoded from the transformed data from $o_{t+1}$, and $z_{t+1}$ is obtained from $P_z(z_{t+1}|z_t, a_t)$, we derive the following Bellman Equations:

$$
\begin{aligned}
\mathcal{U}^\pi Q(s_t, a_t) &= \mathbb{E}_\tau\big[(r_t + \gamma \tilde{r}_{t+1} + \gamma^2 \tilde{r}_{t+2} + \ldots) \mid \pi, s_t\big] \\
&= \mathbb{E}_\tau\big[r_t \mid \pi, s_t\big] + \gamma \, \mathbb{E}_\tau\big[(\tilde{r}_{t+1} + \gamma \tilde{r}_{t+2} + \ldots) \mid \pi, s_t\big] \\
&= r_t + \gamma \, \mathbb{E}_{\tilde{h}_{t+1} \sim P^{\text{Aug}}, z_{t+1} \sim P_z} \mathbb{E}_\tau\big[(\tilde{r}_{t+1} + \ldots) \mid \pi, \tilde{s}_{t+1}\big] \\
&= r_t + \gamma \, \mathbb{E}_{\tilde{h}_{t+1} \sim P^{\text{Aug}}} V^\pi(\tilde{s}_{t+1}),
\end{aligned}
\tag{4}
$$

where $\tilde{r}_t = R(\tilde{h}_{t:t+1}, z_{t:t+1})$, $s_t = (h_t, z_t)$, and $\tilde{s}_t = (\tilde{h}_t, z_t)$. Notably, Eq. 4 is valid only if the transformation from $h_t$ to $\tilde{h}_{t+1}$ is independent of $a_t$ and also independent of the deterministic transitions of the other state branch $P_z(z_{t+1}|z_t, a_t)$. This Bellman Equation supports the feasibility of the proposed TD method in Eq. 3, which computes the TD estimate based on the current-step original data while computing the TD targets based on the next-step transformed data.

## 4 EXPERIMENTS

### 4.1 EXPERIMENTAL SETUP

We evaluate MetaTrader using the following datasets adopted from StockFormer (Gao et al., 2023a):

- *CSI-300 dataset*: This dataset is collected from the CSI-300 Composite Index with 88 stocks. It ranges from 01/17/2011 to 04/01/2022, and is divided into training and test splits with 1,936 and 785 trading days respectively.

- *NASDAQ-100 dataset*: This dataset contains 86 NASDAQ stocks and is collected from Yahoo Finance. It ranges from 01/17/2011 to 04/01/2022, with a training set of 2,002 trading days and a test set of 819 trading days.

On both datasets, we leverage two training and evaluation setups:

- *Offline evaluation*: We conduct in-distribution model finetuning on the last-year data within the training set, *i.e.*, 01/04/2018—12/31/2018. The test period is 04/01/2019—04/01/2022.

- *Online adaptation*: We conduct finetuning on-the-fly over the streaming test data. Specifically, the test set is divided into three equal-length periods. We finetune the model using the previous test split before evaluating it using the next split. Please refer to Appendix B for more details.

We mainly use the following models for comparison:

- *Market benchmarks*, including the CSI-300 Index and the NASDAQ Composite Index.

- *RL trading methods*, including FinRL (Liu et al., 2021), SARL (Ye et al., 2020), and StockFormer (Gao et al., 2023a).

- *Offline RL methods,* including CQL (Kumar et al., 2020) and IQL (Kostrikov et al., 2021).

- *Stock prediction models*, including HATR (Wang et al., 2021), Relational Ranking (Feng et al., 2019), AutoFormer (Wu et al., 2021), and FactorVAE (Duan et al., 2022). We use the *buy-and-hold* strategy for the stock prediction methods, *i.e.*, buying the stock which has the highest estimated return in the next 5 days and selling it 5 days later.

Table 1: Offline evaluation results. We use *cumulative return* (CR), *annualized return* (AR), and *Sharpe ratio* (SR) as evaluation metrics. Please refer to Appendix C for their detailed definitions.

| Method | CSI-300 | | | NASDAQ-100 | | |
|---|---|---|---|---|---|---|
| | $CR^{\uparrow}$ | $AR^{\uparrow}$ | $SR^{\uparrow}$ | $CR^{\uparrow}$ | $AR^{\uparrow}$ | $SR^{\uparrow}$ |
| Market benchmark | 0.08 | 0.02 | 0.23 | 0.99 | 0.26 | 0.98 |
| HATR | -0.05 | -0.02 | 0.06 | 0.10 | 0.03 | 0.25 |
| Relational Ranking | -0.13 | -0.05 | -0.05 | 0.79 | 0.22 | 0.75 |
| AutoFormer | -0.08 | -0.03 | 0.02 | -0.28 | -0.10 | -0.27 |
| FactorVAE | 0.96 | 0.25 | 1.25 | 0.90 | 0.24 | 0.77 |
| SARL | 1.06±0.14 | 0.27±0.03 | 0.98±0.08 | 1.03±0.20 | 0.27±0.04 | 0.80±0.09 |
| CQL | 0.64±0.07 | 0.18±0.02 | 0.75±0.05 | 0.77±0.12 | 0.21±0.02 | 0.76±0.06 |
| IQL | 1.02±0.10 | 0.26±0.02 | 0.94±0.06 | 0.92±0.09 | 0.24±0.02 | 0.87±0.04 |
| FinRL-SAC | 0.83±0.05 | 0.22±0.01 | 0.92±0.04 | 0.37±0.05 | 0.11±0.01 | 0.54±0.04 |
| FinRL-DDPG | 0.58±0.15 | 0.16±0.04 | 0.73±0.12 | 0.91±0.11 | 0.24±0.02 | 0.75±0.05 |
| StockFormer | 1.24±0.10 | 0.31±0.02 | 1.20±0.06 | 0.98±0.07 | 0.26±0.02 | 0.93±0.04 |
| MetaTrader *w/o* finetune | 1.27±0.08 | 0.31±0.02 | 1.21±0.05 | 1.08±0.07 | 0.28±0.02 | 0.92 ± 0.05 |
| MetaTrader | **1.44±0.07** | **0.35±0.02** | **1.35±0.08** | **1.30±0.08** | **0.32±0.02** | **1.11±0.04** |

Table 2: Online adaptation results. We divide the entire test set into three equal-length splits and progressively finetune the models throughout the streaming test set.

| Method | CSI-300 | | | NASDAQ-100 | | |
|---|---|---|---|---|---|---|
| | $CR^{\uparrow}$ | $AR^{\uparrow}$ | $SR^{\uparrow}$ | $CR^{\uparrow}$ | $AR^{\uparrow}$ | $SR^{\uparrow}$ |
| Market benchmark | 0.08 | 0.02 | 0.23 | 0.99 | 0.26 | 0.98 |
| FactorVAE-Finetune | 1.07 | 0.27 | 1.32 | 1.02 | 0.26 | 0.84 |
| StockFormer-Finetune | 1.46±0.05 | 0.35±0.01 | 1.37±0.05 | 1.26±0.08 | 0.31±0.02 | 1.03±0.09 |
| MetaTrader | **1.84±0.03** | **0.42±0.01** | **1.61±0.03** | **1.58±0.03** | **0.37±0.01** | **1.47±0.04** |

For the online adaptation setup, our main comparison is between MetaTrader, *FactorVAE-Finetune*, and *StockFormer-Finetune*, which are also continuously finetuned over the streaming test data. All compared models are experimented with market transaction costs. Unless otherwise specified, the results of the RL methods are averaged across three random training seeds. Additionally, please refer to Appendix D for the details of the training hyperparameters.

## 4.2 MAIN RESULTS

**Offline evaluation results.** Table 1 present the quantitative results of MetaTrader for *offline evaluation*. MetaTrader outperforms all stock prediction and RL methods in both cumulative return and Sharpe ratio. In particular, it outperforms FactorVAE by **50%** (1.44 vs. 0.96) in CR and by **8%** (1.35 vs. 1.25) in SR on the CSI dataset, and by **44.4%** (1.30 vs. 0.90) and **44.1%** (1.11 vs. 0.77) on the NASDAQ dataset. As indicated by the investment risk metric, namely the Sharpe ratio, the RL methods tend to make more profitable but riskier investments than the stock prediction models. This is achieved by employing bilevel policy learning, which prevents the policy from overfitting to the offline data. We also evaluate the common techniques to improve the robustness of offline RL agents in out-of-distribution data. We find that MetaTrader outperforms CQL and IQL by large margins.

**Online adaptation results.** Figure 5 and Table 2 present the quantitative comparisons under the *online adaptation* setup, in which we continuously finetune all compared models on the streaming test data. As we can see, MetaTrader presents a remarkable advantage against other approaches, including the state-of-the-art stock prediction model (*i.e.*, FactorVAE) and RL-based stock trading method (*i.e.*, StockFormer). On the CSI dataset, it improves StockFormer-Finetune by **26%** (1.46 → 1.84) in cumulative return and by around **18%** (1.37 → 1.61) in Sharpe ratio. On the NASDAQ dataset, MetaTrader improves StockFormer-Finetune by over **25%** (1.26 → 1.58) in cumulative return and by around **43%** (1.03 → 1.47) in Sharpe ratio. In conclusion, MetaTrader performs well in online adaptation, which aligns with real trading scenarios.

## 4.3 MODEL ANALYSES

**The effectiveness of data transformation.** To assess the true impact of various data transformation techniques proposed in Section 3.2, we experiment with baseline models that (i) do not incorporate

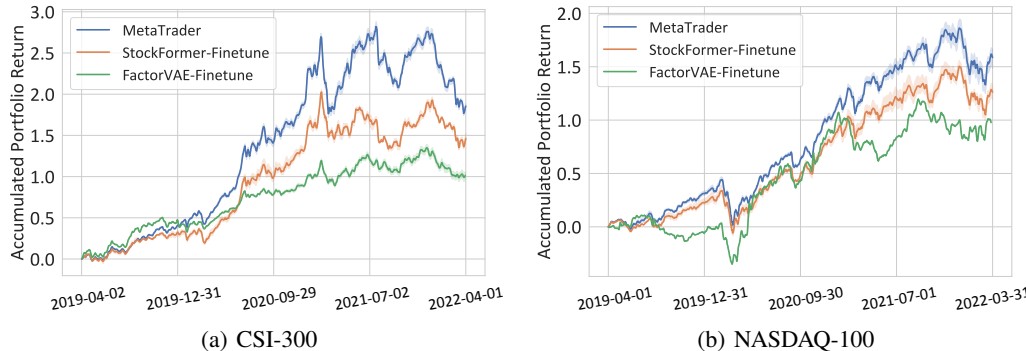

| (a) CSI-300 | (b) NASDAQ-100 |

Figure 5: The accumulated returns under the online adaptation setup with ten random seeds.

Table 3: Analyses of data transformation techniques used in the *out-of-distribution policy learning* phase (Alg. 1). We report the mean results on CSI-300 over three seeds. DT: data transformation

| Method | CR$^\uparrow$ | AR$^\uparrow$ | SR$^\uparrow$ | Method | CR$^\uparrow$ | AR$^\uparrow$ | SR$^\uparrow$ |
|---|---|---|---|---|---|---|---|
| *w/o* DT | 1.66±0.03 | 0.385±0.007 | 1.44±0.02 | $F_3$ | 1.73±0.03 | 0.398±0.008 | 1.53±0.04 |
| $F_1$ | 1.69±0.03 | 0.390±0.008 | 1.53±0.03 | $F_1 + F_2$ | 1.77±0.02 | 0.404±0.006 | 1.59±0.03 |
| $F_2$ | 1.67±0.03 | 0.389±0.008 | 1.50±0.03 | $F_1 + F_2 + F_3$ | **1.84±0.03** | **0.417±0.008** | **1.61±0.03** |

transformed data in any training phases, and (ii) the ones that only incorporate parts of the data transformation techniques. We have two observations from Table 3. First, leveraging **ANY** of the data transformation methods in the OOD policy learning phase consistently proves beneficial for the model's final performance, leading to significant improvements across all three metrics. Second, using a combination of various transformation techniques results in significant improvements. There is a notable $10.8\%$ ($1.66 \rightarrow 1.84$) increase in cumulative return for online adaptation on CSI-300.

**Impact of the ensemble-based conservative TD method.** To assess the effectiveness of the ensemble-based TD method detailed in Section 3.3, we implement a baseline model of MetaTrader that employs the original TD method from SAC. Table 4 demonstrates the improvements achieved by our proposed TD method, with a significant increase of $9.5\%$ in cumulative return on the CSI-300 dataset and a $6.8\%$ increase on NASDAQ-100.

**Can our TD method alleviate value overestimation?** We compare the value estimation results with vs. without the ensemble-based TD method. In Figure 6, we report the discrepancies between the values predicted by the critic models and true values, determined by the discounted sum of rewards throughout the same

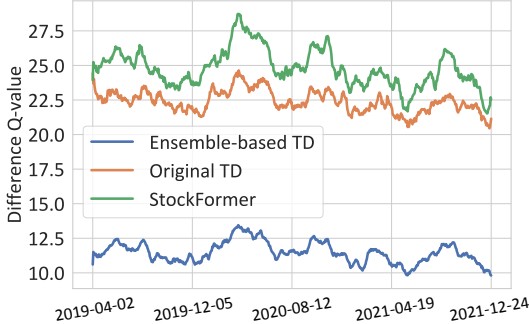

Figure 6: The disparities between the predicted values by the critic and the true discounted returns. A larger disparity indicates a more pronounced value overestimation. The results are obtained under the offline evaluation setup on the CSI-300 dataset.

data trajectories. As observed, StockFormer and "*MetaTrader w/ original TD*" tend to overestimate the true value function. In contrast, the values estimated by the final "*MetaTrader w/ ensemble-based TD*" are notably more accurate and more akin to the true values.

**Technical designs in model finetuning.** In Alg. 2, we conduct model finetuning on real data from the recent year, which is close to the testing period, using bilevel gradient updates. First, by comparing "*MetaTrader w/o finetune*" with the final MetaTrader from Table 1, we note a decline in performance without the finetuning phase. It is essential to highlight that the finetuning data is also included within the dataset during the OOD policy learning process. Furthermore, we explore the necessity of bilevel optimization and demonstrate why stock transformation is not used during the in-distribution finetuning phase. As shown in Table 5, compared with directly using the inner-loop gradients to update the model, leveraging bilevel optimization leads to a $3.4\%$ improvement in the cumulative return on CSI-300 ($1.84$ vs. $1.78$) and a $12.1\%$ improvement on NASDAQ-100 ($1.58$ vs. $1.41$). Moreover, in Table 5, we can see that incorporating data transformation in the finetuning phase

Table 4: Ablation studies of the ensemble-based TD target in the *out-of-distribution policy learning* phase (Alg. 1). The experiments are conducted under the online adaptation setup.

| Method | CSI-300 | | | NASDAQ-100 | | |
|---|---|---|---|---|---|---|
| | $CR^{\uparrow}$ | $AR^{\uparrow}$ | $SR^{\uparrow}$ | $CR^{\uparrow}$ | $AR^{\uparrow}$ | $SR^{\uparrow}$ |
| Original | 1.68±0.04 | 0.39±0.01 | 1.49±0.03 | 1.48±0.03 | 0.35±0.01 | 1.33±0.04 |
| Ensemble-based | **1.84**±0.03 | **0.42**±0.01 | **1.61**±0.03 | **1.58**±0.03 | **0.37**±0.01 | **1.47**±0.04 |

Table 5: Ablation studies of the operations in the *in-distribution model finetuning* phase (Alg. 2), including the learning scheme with bilevel gradient update and the use of transformed stock data. The experiments are conducted under the online adaptation setup.

| Bilevel optimization | Transformed data | CSI-300 | | | NASDAQ-100 | | |
|---|---|---|---|---|---|---|---|
| | | $CR^{\uparrow}$ | $AR^{\uparrow}$ | $SR^{\uparrow}$ | $CR^{\uparrow}$ | $AR^{\uparrow}$ | $SR^{\uparrow}$ |
| ✗ | ✗ | 1.78±0.03 | 0.41±0.01 | 1.57±0.03 | 1.41±0.04 | 0.34±0.01 | 1.34±0.04 |
| ✓ | ✗ | **1.84**±0.03 | **0.42**±0.01 | **1.61**±0.03 | **1.58**±0.03 | **0.37**±0.01 | **1.47**±0.04 |
| ✓ | ✓ | 0.84±0.04 | 0.23±0.01 | 0.94±0.05 | 1.24±0.03 | 0.31±0.01 | 1.03±0.05 |

leads to a clear decline in performance. This is reasonable, as the transformed data may not align with the recent dynamics patterns close to the test set.

**Additional gradient steps for the baselines.** As our model is optimized for $30k$ steps during OOD policy learning and for $5k$ steps during model finetuning, we increase the training steps of other compared models to $35k \times 2$ and $35k \times 2 \times K$ steps respectively, where $K$ corresponds to the number of sampled subsets in each bilevel optimization step in our method. We can see from Table 6 that after convergence, continuing training does not yield significant improvements for the baseline models.

**Computational costs.** In Table 7, we present the total training time and the per-sequence inference time of the compared models on a single NVIDIA RTX 3090 GPU. Given that our work primarily focuses on daily-level stock trading, the increased training cost introduced by bilevel optimization is acceptable, while the inference time adequately meets the efficiency demands in this scenario.

### 4.4 Challenges in Handling Larger Market Data

Existing RL-based stock trading methods, such as FinRL, StockFormer, and SARL, primarily conduct experiments on relatively small-scale datasets. We attribute this limitation to two main factors. From a data perspective, trading suspensions frequently occur in real-world stock data. Previous studies often select stocks based on the requirement that the proportion of valid data exceeds a specific threshold (*e.g.*, 98% in StockFormer) to reduce noise from excessive data interpolation.

From an algorithm perspective, as the stock pool size increases, the action space grows significantly, making it more challenging for RL methods to manage. If we aim to trade thousands of stocks in the market, the dimensionality of the action space can be even larger than the number of training sequences. The difficulty of high-dimensional action space is well-documented in other domains beyond stock trading (Tavakoli et al., 2018; Saito et al., 2024).

Despite these challenges, we provide experimental results on a larger stock market in Appendix F.1.

## 5 Related Work

There are two primary groups of deep learning-based approaches for portfolio optimization.

The first one leverages the temporal modeling capabilities of existing models to make future predictions of stock prices (Li et al., 2018; Xu & Cohen, 2018; Feng et al., 2019; Wang et al., 2021; Duan et al., 2022; Zheng et al., 2023). For stock trading, these methods are usually combined with a relatively simple trading policy (such as buying stocks predicted to have the highest gains and selling them at a set time). The second line of work is based on deep reinforcement learning that frames portfolio optimization as MDPs and makes dynamic decisions on the timing and quantity of the investment (Deng et al., 2016; Briola et al., 2021; Jeong & Kim, 2019; Liu et al., 2021; Kumar, 2023; Liu et al., 2022; Gao et al., 2023a). Still, previous attempts have shown that policies, limited by offline state exploration, tend to remember only the optimal policy from offline data, reducing flexibility and generalizability. Although our method under the online adaptation has a similar training setup

Table 6: Results of the compared models with a larger number of optimization steps. The results are obtained on CSI-300 under the offline evaluation setup over three random seeds.

| Method | Optim. steps | $CR^\uparrow$ | $AR^\uparrow$ | $SR^\uparrow$ | Optim. steps | $CR^\uparrow$ | $AR^\uparrow$ | $SR^\uparrow$ |
|---|---|---|---|---|---|---|---|---|
| SARL | $35k \times 2$ | 1.01 | 0.26 | 0.95 | $35k \times 64$ | 1.04 | 0.27 | 0.99 |
| FinRL-SAC | $35k \times 2$ | 0.86 | 0.23 | 0.94 | $35k \times 64$ | 0.89 | 0.24 | 0.93 |
| FinRL-DDPG | $35k \times 2$ | 0.63 | 0.18 | 0.77 | $35k \times 64$ | 0.65 | 0.18 | 0.79 |
| StockFormer | $35k \times 2$ | 1.26 | 0.31 | 1.21 | $35k \times 64$ | 1.28 | 0.32 | 1.24 |
| MetaTrader | $35k$ | **1.44** | **0.35** | **1.35** | - | - | - | - |

Table 7: Computational cost.

| Method | Training time | Inference time per sequence |
|---|---|---|
| StockFormer *w/* pretrained feature extractors | 28min 02s | 19.03ms |
| StockFormer from scratch | 55min 03s | 19.03ms |
| MetaTrader *w/* pretrained feature extractors | 37min 27s | 19.06ms |
| MetaTrader from scratch | 64min 28s | 19.06ms |

to offline-to-online RL (Nair et al., 2020; Zhang et al., 2023; Yu & Zhang, 2023; Zhao et al., 2023), which is mainly designed to address the high cost of online training, we aim to learn a generalization strategy under diverse market conditions.

Another group of existing methods related to MetaTrader is the bilevel optimization-based meta-learning, which has been widely used in few-shot learning (Antoniou et al., 2019; Li et al., 2019; Triantafillou et al., 2020; Day et al., 2022; Cheng et al., 2023) and domain adaptation (Schmidhuber, 1987; Finn, 2018; Hospedales et al., 2021). In the realm of RL, it has been employed for learning dynamics models (Sæmundsson et al., 2018; Nagabandi et al., 2019) or directly learning the policies (Duan et al., 2017; Mishra et al., 2018; Finn et al., 2017; Nagabandi et al., 2019; Gupta et al., 2018; Humplik et al., 2019; Mitchell et al., 2021; Pong et al., 2022; Tang, 2022; Greenberg et al., 2023; Gao et al., 2023b; Ma et al., 2023; Wang et al., 2023). These models have already demonstrated the potential of meta-learning to enhance the generalizability of the RL policy. Unlike previous work, we specifically tackle the challenges of policy learning with limited and non-stationary financial data. Accordingly, we propose a new bilevel RL approach to improve the policy's generalizability and alleviate the value overestimation issue as well.

## 6 CONCLUSIONS AND LIMITATIONS

This paper presents MetaTrader, an RL method that formulates stock trading as an offline RL problem with decoupled MDPs. MetaTrader improves the model's generalizability to non-stationary stock data by integrating carefully designed stock augmentation techniques in a bilevel policy learning framework. Additionally, we proposed a novel TD method with an ensemble-based TD target, which aims to produce more conservative policies in scenarios with limited data. Experiments on two public stock datasets demonstrate the effectiveness of MetaTrader compared to existing RL-for-finance approaches, showcasing its great potential in dealing with rapidly changing market data.

Our approach is trained and validated on daily-level stock trading data, and its effectiveness has been demonstrated across two datasets through extensive experiments. With an execution time per inference of approximately 20 milliseconds, our method shows potential applicability in high-frequency trading scenarios. Moreover, the proposed framework, which initially incorporates specific data transformation techniques to enhance the datasets and subsequently employs bilevel reinforcement learning with an ensemble-based TD target, can be considered as a general technique suitable for various decision-making problems in time series information systems, such as energy load forecasting, traffic flow management, and healthcare monitoring.

An unresolved problem in this study is the stability of reinforcement learning. In experiments, we noted that RL-based methods (including SARL, StockFormer, and our approach) typically exhibit larger standard deviations in performance across multiple training runs with random seeds, compared to stock prediction methods (*e.g.*, FactorVAE and HATR). This phenomenon is a common difficulty for the current state of research within this field. To alleviate this issue, we plan to explore robust reinforcement learning techniques in the future.

## ETHICS STATEMENT

By combining bilevel optimization with reinforcement learning on non-stationary stock data, our study paves the way for developing intelligent trading agents that can adapt and learn from limited financial data, improving their decision-making abilities in rapidly changing market conditions. This advancement is crucial in empowering asset managers and individual investors to make data-driven decisions that effectively respond to the evolving dynamics of the market. One potential negative social impact of learning-based stock trading methods is increased economic inequality, especially when advanced trading strategies are predominantly available to giant institutional investors. Individual investors might face challenges in competing on an equal footing, potentially limiting their ability to benefit from financial markets. Addressing this concern involves promoting inclusive access to the technologies and ensuring that advancements in machine learning benefit a broad spectrum of market participants.

## REPRODUCIBILITY STATEMENT

We prioritize the reproducibility of our work. All results can be reproduced by following the experimental details presented in Section 4 and Appendix A. We also report all hyperparameters involved in our method in Appendix D. We will release the code upon paper acceptance.

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

## A    DETAILS OF STOCK DATA TRANSFORMATION

As outlined in Section 3.2, we transform the data in sequences of $64$ days in length to construct the subsets. While the data transformation techniques are briefly illustrated in the main text with Figure 4, we here present more descriptions of the implementation details.

Consider a specific stock $A$: It provides an input sequence to the model, in which the daily closing prices can be denoted as $O_{F_0}^{\text{close}} = \{o_0^{\text{close}}, o_1^{\text{close}}, \ldots, o_{63}^{\text{close}}\}$. The subsequent prices after this sequence are $o_{64}^{\text{close}}, o_{65}^{\text{close}}, \ldots$, and so forth. Accordingly, we have the sequence of growth rate between daily closing prices for this stock: $\Delta O_{F_0}^{\text{close}} = \{0, \Delta o_1^{\text{close}}, \ldots, \Delta o_{63}^{\text{close}}\}$. For example, $o_9^{\text{close}} = o_8^{\text{close}} \times (1 + \Delta o_9^{\text{close}})$. Without loss of generality, let us assume that its daily growth rate on the 10-th day (*i.e.*, $\Delta o_9^{\text{close}}$) is among the Top-$10\%$ within the stock pool.

In the first transformation method, the price sequence of stock $A$ is transformed into another sequence denoted by $O_{F_1}^{\text{close}} = \{o_0^{\text{close}}, o_1^{\text{close}}, \ldots, o_8^{\text{close}}, o_9', o_{10}', o_{11}', \ldots, o_{63}'\}$, where $o_9' = o_8^{\text{close}} \times (1 - \Delta o_9^{\text{close}})$. We retain the daily price growth rates on other days, such that $\Delta o_t' = \Delta o_t^{\text{price}}$ for $t \geq 10$. In particular, on days when the number of stocks with positive price growth does not reach $10\%$ of the total, only those stocks with positive growth will have inverted growth rates. It is noteworthy that although we manipulate the input data by measuring the daily closing prices only, we also transform the open/high/low prices along with the closing prices, while keeping the original data for the trading volumes unchanged.

In the second transformation method, the original price sequence of stock $A$ is reversed to construct another sequence of $O_{F_2} = \{o_{63}^{\text{close}}, o_{62}^{\text{close}}, \ldots, o_0^{\text{close}}\}$.

In the third transformation method, the price sequence is transformed into $O_{F_3} = \{o_0^{\text{close}}, o_4^{\text{close}}, o_8^{\text{close}}, \ldots, o_{248}^{\text{close}}, o_{252}^{\text{close}}\}$.

For transformations $F_2$ and $F_3$, all input data (high/low/volume) will be shifted alongside corresponding price data. For all data transformation methods, we carefully divided the training and test sets based on dates, ensuring that all transformations were applied exclusively to the training set. This guarantees no data leakage and ensures a fair comparison among all methods.

## B    DATASETS

### B.1    CSI-300 STOCK DATASET

We follow previous work (Feng et al., 2019; Gao et al., 2023a) to retain the stocks that have been traded on more than $98\%$ training days since 01/17/2011. If a stock is suspended from trading, we interpolate the missing training data using the daily changing rate of CSI-300 Composite Index.

For online adaptation, a portion of the test set data is used for in-distribution model finetuning. The entire test set is divided into three equal periods, each followed by in-distribution model finetuning before testing. Specifically, for the test period from 04/01/2019 to 04/01/2020, the testing approach is equivalent to that of the offline setting. For the period from 04/02/2020 to 04/01/2021, we conduct in-distribution model finetuning using real data from 04/01/2019 to 04/01/2020 before testing. Similarly, for the test period from 04/01/2021 to 04/02/2022, we conduct in-distribution model finetuning using real data from 04/02/2020 to 04/01/2021 before testing.

For offline evaluation, we exclusively use the training set for both training and finetuning, and evaluate the model on the entire test set spanning three years. Specifically, we conduct inner loop optimization and outer loop optimization with stock transformations using training data from 01/17/2011 to 12/31/2018, and then conduct in-distribution model finetuning using real stock data from 01/04/2018 to 12/31/2018. The model is then evaluated on the complete test set.

### B.2    NASDAQ-100 STOCK DATASET

Like in CSI-300, we use the $98\%$ criteria to filter stocks, which derives an investment pool of $86$ stocks and then fill in the missing data based on the daily rate of change of the NASDAQ 100 Index.

We employ both online and offline evaluation setups. For online adaptation, similar to that on the CSI-300 stock dataset, the entire test set is divided into three equal periods, each followed by in-distribution model finetuning before testing.

For offline evaluation, we conduct inner loop optimization and outer loop optimization with stock augmentations using training data from 01/17/2011 to 12/31/2018. We conduct in-distribution model finetuning using real stock data from 01/02/2018 to 12/31/2018.

### B.3 DATA NORMALIZATION

We perform normalization separately for each stock, ensuring that all normalization factors are specific to the data of the individual stock. For a given stock, all price data (open, close, high, low) share the same normalization factor. The normalized values can be formulated as

$$N_{t_i}^{\text{price}} = \frac{o_{t_i}^{\text{price}} - \min_t\{o_t^{\text{low}}\}}{\max_t\{o_t^{\text{high}}\} - \min_t\{o_t^{\text{low}}\}}, \tag{5}$$

The normalization for volume is expressed as

$$N_{t_i}^{\text{volume}} = \frac{o_{t_i}^{\text{volume}} - \min_t\{o_t^{\text{volume}}\}}{\max_t\{o_t^{\text{volume}}\} - \min_t\{o_t^{\text{volume}}\}}, \tag{6}$$

where $N_{t_i}$ represents the normalized value, and the superscript "price" refers to the four price data types: open, close, high, and low.

## C METRICS

**Cumulative return (CR):** This is a measure of the income generated by an investment portfolio over a specific period. Specifically, it includes the entire test period.

$$o_t^{\text{close}} \in \mathbb{R}^{|S|}, \quad z_t' = z_t^{(2:|S|+1)} \in \mathbb{R}^{|S|}$$

$$A_t = z_t' \cdot o_t^{\text{close}} = \sum_{i=1}^{|S|} z_t'^{(i)} \cdot o_t^{\text{close}\,(i)}, \quad CR_t = A_t/A_0 - 1 \tag{7}$$

where $A_t$ represents the total asset value at time $t$ and $A_0$ denotes the initial asset value. In practice, we assume all transactions are executed at the closing price $o_t^{\text{close}}$.

**Annualized return (AR):** This is a measure of the investment growth over one year.

$$AR = CR_t^{\frac{d}{t}} - 1, \tag{8}$$

where $d$ represents the total number of trading days in one year.

**Sharpe ratio (SR):** This is a metric in finance to measure the performance of an investment compared to a risk-free asset.

$$SR = \frac{CR - R_f}{\sigma_p}, \tag{9}$$

where $R_f$ is the risk-free rate of return. $\sigma_p$ is the standard deviation of the portfolio's excess return. For our experiments, the risk-free rate used in the analysis is set to 0.

## D HYPERPARAMETERS

In Table 8, we provide the hyperparameter details in both the OOD policy learning phase and the in-distribution model finetuning phase. For the feature extraction module, we adopt the identical hyperparameters as those employed in StockFormer (Gao et al., 2023a).

Table 8: Hyperparameters in the OOD policy learning phase and in-distribution finetuning phase.

| Notation | Hyperparameter | Description |
|---|---|---|
| $\eta_1$ | 0.00001 | learning rate of the critic (inner loop) |
| $\eta_2$ | 0.0001 | learning rate of the critic (outer loop) |
| $\alpha_1$ | 0.00001 | learning rate of the actor (inner loop) |
| $\alpha_2$ | 0.0001 | learning rate of the actor (outer loop) |
| $d_{\text{hidden}}^1$ | 256 | number of MLP channels in the critic |
| $d_{\text{hidden}}^2$ | 256 | number of MLP channels in the actor |
| $B, K$ | 32 | batch size, number of sampled subsets per iteration |
| $M$ | 216 | number of time period slices |
| $N$ | 3 | number of stock augmentation techniques |
| $L$ | 64 | length of time period slices |

Table 9: Technical Indicators and Descriptions

| Technical Indicator | Description |
|---|---|
| macd | Moving average convergence divergence |
| boll_ub | Bollinger bands (upper band) |
| boll_lb | Bollinger bands (lower band) |
| rsi_30 | 30 periods relative strength index |
| cci_30 | Retrieves the 30 periods commodity channel index |
| dx_30 | Directional index with a window length of 30 |
| close_30_sma | 30 periods simple moving average of the close price |
| close_60_sma | 60 periods simple moving average of the close price |

## E    TECHNICAL INDICATORS

The technical indicators mentioned in the paper follow the settings used in StockFormer (Gao et al., 2023a). Specifically, we use the Stockstats package for data analysis. The technical indicators employed are listed in Table 9.

## F    ADDITIONAL RESULTS

### F.1    EXPERIMENTAL RESULTS ON LARGER DATASET

We conduct experiments on a larger dataset by expanding the range of CSI stocks and selecting a dataset containing 587 stocks. We maintain the same experimental setup as in the offline evaluation and compare our method with several baselines. The results are presented in Table 10.

### F.2    RISK EVALUATION BY MAXIMUM DRAWDOWN

In stock trading tasks, achieving high returns should be balanced with risk management. Therefore, we introduce the maximum drawdown (MDD) metric to evaluate the investment risk of each method, providing a more comprehensive assessment of their performance, as shown in Table 11.

Table 10: Offline evaluation results on the expanded dataset with 587 stocks.

| Method | CR$^{\uparrow}$ | AR$^{\uparrow}$ | SR$^{\uparrow}$ | MDD$^{\downarrow}$ |
|---|---|---|---|---|
| Market benchmark | 0.15 | 0.05 | 0.30 | 0.29 |
| SARL | 0.16 | 0.05 | 0.28 | 0.47 |
| FinRL-SAC | -0.12 | -0.04 | -0.03 | 0.44 |
| StockFormer | 0.18 | 0.06 | 0.32 | 0.39 |
| MetaTrader | **0.41** | **0.12** | **0.60** | **0.37** |

Table 11: Maximum drawdown (MDD) results of the offline evaluation

| Dataset | Market benchmark | HATR | SARL | FinRL-SAC | StockFormer | Metatrader |
|---|---|---|---|---|---|---|
| CSI-300 | 0.31 | 0.51 | 0.36±0.02 | 0.30±0.01 | 0.31±0.02 | 0.28±0.02 |
| NASDAQ-100 | 0.28 | 0.35 | 0.40±0.01 | 0.32±0.01 | 0.32±0.02 | 0.31±0.00 |

Table 12: Offline evaluation results on more recent data.

| Method | CR$^{\uparrow}$ | AR$^{\uparrow}$ | SR$^{\uparrow}$ | MDD$^{\downarrow}$ |
|---|---|---|---|---|
| Market benchmark | -0.08 | -0.04 | 0.02 | 0.32 |
| SARL | -0.13 | -0.07 | -0.07 | 0.51 |
| FinRL-SAC | 0.04 | 0.01 | 0.03 | 0.49 |
| StockFormer | 0.21 | 0.10 | 0.46 | 0.45 |
| MetaTrader | **0.32** | **0.15** | **0.76** | **0.44** |

It can be observed that our method performs the best in terms of the MDD metric among all reinforcement learning methods. This suggests that our method can learn more robust and high-yield policies to a certain extent.

### F.3 EVALUATION ON MORE RECENT DATA

We used data up to 2022 to ensure a fair comparison with StockFormer (Gao et al., 2023a), which follows the same training and testing period division. Moreover, we conduct additional experiments using data beyond 2022. In this experiment, we do not extend the training set range but directly test on the CSI-300 dataset spanning from 2022-05-01 to 2024-05-01. As shown in Table 12, during this period, the overall market is weaker than that in the original test set before 2022. Consequently, the annualized returns of all methods are reduced. Nonetheless, our method consistently outperforms all baselines, highlighting its potential for profitability even under more challenging market conditions.

### F.4 THE EFFECTIVENESS OF FINETUNING OF RL-FOR-FINANCE MODELS

In practical RL-for-finance tasks, the naive fine-tuning approach often fails to enhance model performance on test data. This is primarily due to overfitting to specific data patterns when finetuning on more recent data. This is precisely why we propose the bilevel optimization approach for the RL method. Theoretically, the bilevel optimization scheme can significantly enhance the model's generalizability to new data. Similar approaches, known as model-agnostic meta-learning (MAML) (Finn et al., 2017), have been widely adopted to improve finetuning results in few-shot learning scenarios. Intuitively, it aims to find well-performed parameter initialization that can be quickly adapted to a new related task using only a few data and a few gradient steps.

We compare the performance of different RL methods with and without finetuning, using the same configuration as offline evaluation. We present the CR, PR, SR, and MDD on the CSI-300 dataset in Table 13. The results are averaged over three random training seeds. Notably in the cumulative return metric, our bilevel optimization approach significantly improves the finetuning results (by $+13.39\%$), while the previous RL approaches do not support such effective model finetuning (*e.g.*, by $+0.81\%$ for StockFormer).

Table 13: A comparison on whether finetuning is performed. We use *cumulative return* (CR), *annualized return* (AR), and *Sharpe ratio* (SR) as evaluation metrics.

| Method | Finetune | | | Train from scratch | | |
|---|---|---|---|---|---|---|
| | CR$^\uparrow$ | AR$^\uparrow$ | SR$^\uparrow$ | CR$^\uparrow$ | AR$^\uparrow$ | SR$^\uparrow$ |
| SARL | 1.06±0.14 | 0.27±0.03 | 0.98±0.08 | 1.03±0.13 | 0.27±0.03 | 0.89±0.08 |
| CQL | 0.64±0.07 | 0.18±0.02 | 0.75±0.05 | 0.69±0.05 | 0.19±0.01 | 0.83±0.05 |
| IQL | 1.02±0.10 | 0.26±0.02 | 0.94±0.06 | 0.96±0.10 | 0.25±0.02 | 0.89±0.04 |
| FinRL-SAC | 0.83±0.05 | 0.22±0.01 | 0.92±0.04 | 0.80±0.07 | 0.22±0.02 | 0.82±0.05 |
| FinRL-DDPG | 0.58±0.15 | 0.16±0.04 | 0.73±0.12 | 0.63±0.13 | 0.18±0.04 | 0.77±0.09 |
| StockFormer | 1.24±0.10 | 0.31±0.02 | 1.20±0.06 | 1.23±0.09 | 0.31±0.02 | 1.18±0.05 |
| MetaTrader | **1.44**±0.07 | **0.35**±0.02 | **1.35**±0.08 | 1.27±0.08 | 0.31±0.02 | 1.21±0.05 |

Table 14: Offline evaluation results on different ensemble method.

| Method | CR$^\uparrow$ | AR$^\uparrow$ | SR$^\uparrow$ | MDD$^\downarrow$ |
|---|---|---|---|---|
| Minimum value | 1.17 | 0.30 | 1.03 | 0.34 |
| Mean value | 1.10 | 0.28 | 0.99 | 0.33 |
| Ours | **1.44** | **0.35** | **1.35** | **0.28** |

## G   THE COMPARISON WITH EXISTING ENSEMBLE Q-LEARNING METHODS

The key differences between our conservative TD method and other ensemble-based Q-learning methods can be summarized as follows:

- Existing methods are based on **model diversity**: Most existing ensemble-based methods require multiple Q-networks with identical input data $(s_{t+1}, a_{t+1})$ to calculate a conservative TD target:

$$\hat{Q}'(s_t, a_t) = r_t + \gamma \big[ - \lambda \log \pi_\theta(\hat{a}_{t+1} \mid s_{t+1}) + \underset{k=1,\ldots,M}{\Psi} Q_{\bar{\phi}_k}(s_{t+1}, \hat{a}_{t+1}) \big], \qquad (10)$$

  which significantly increases the model size. For example:
    - An et al. (2021) uses the minimum value of multiple parallel Q-networks as the Bellman target;
    - Lee et al. (2022) stabilizes Q-learning by averaging previously learned Q-values as the target;
    - Wu et al. (2022) averages all Q-values, excluding those with the highest $N - K$ values.

- Our method is based on **data diversity**: Our ensemble method is based on original stock data and its transformations $(s_{t+1}^{(n)}, a_{t+1}^{(n)})$ to calculate a conservative TD target by a single Q-function:

$$\hat{Q}'(s_t, a_t) = r_t + \gamma \big[ - \lambda \log \pi_\theta(\hat{a}_{t+1} \mid s_{t+1}) + \min_{k=1,2} \min_{n=1:N} \big( Q_{\bar{\phi}_k}(s_{t+1}, \hat{a}_{t+1}), Q_{\bar{\phi}_k}(s_{t+1}^{(n)}, \hat{a}_{t+1}^{(n)}) \big) \big].$$
$$(11)$$

  This approach leverages transformations of stock data to account for diverse market conditions, thereby capturing more variability in the decision-making process. We illustrate the feasibility of our approach from the perspective of Bellman Equations within our offline RL setup in Eq. (4).

We conduct the experiments by replacing the ensemble method in our model with the methods used in An et al. (2021); Lee et al. (2022), *i.e.*, using the minimum and mean value of five parallel Q-networks as the Bellman target. As shown in Table 14, our method presents a remarkable advantage against other methods.

