# OpenReview forum: "Bilevel Reinforcement Learning for Stock Data with A Conservative TD Ensemble"
_ICLR.cc/2025/Conference — Submitted to ICLR 2025_

### Official Review · Reviewer_jZAh · 2024-10-21

**Soundness:** 3
**Presentation:** 3
**Contribution:** 2
**Rating:** 5
**Confidence:** 4

**Summary:**

This paper seeks to enhance classical reinforcement learning through the introduction of two techniques: bilevel learning and conservative TD ensemble. The effectiveness of these improvements is evaluated using experiments on two stock datasets.

**Strengths:**

The paper is well-structured, and the ablation studies are conducted thoroughly.

**Weaknesses:**

### Major Issues:

1. **Lack of Discussion on Non-Stationarity**: There is insufficient discussion on how the paper addresses the issue of non-stationarity. As I understand it, the paper attempts to mitigate non-stationarity through specific data transformations. However, the rationale behind the choice of these three particular transformations is unclear. For instance, in F1, why is only the top 10% considered and not the bottom 10% as well? The action space allows for both long and short positions. Regarding F2, how does disrupting the order of observations in time help overcome non-stationarity? If time observations are treated as completely independent, the sequence would be unpredictable. For F3, as explained in Appendix A, future data is compressed into the present to create a non-empty input. Applying this approach in the out-of-sample leads to the future information issue. These questions raise concerns about the experimental validity.

2. **Input Construction and Covariance Matrix**: A critical component of the model input is the sample covariance matrix between stocks. How exactly is this matrix computed? According to random matrix theory, when the number of assets is close to the number of time observations, the sample covariance matrix becomes singular and provides a biased estimate of the true covariance matrix. Although this method may still be applicable to the two datasets in the paper (with 300 and 100 stocks, respectively), it raises concerns about the scalability of this approach to larger datasets.

3. **Experimental Shortcomings**:

  - First, lines 401-412 involve the calculation of true values, and there appear to be issues with this in the paper. To compute the true value theoretically, a globally optimal policy is required. I may have missed some details, but I hope the authors can clarify this in their rebuttal.
  - Second, regarding the performance metrics, the portfolio return aligns with the annual return. It would strengthen the results if the maximum drawdown (MDD) were also reported alongside the annual return.
  - Finally, the stock prediction strategy—buying only the single stock with the highest predicted value—is problematic, as it introduces excessive randomness into the strategy. A more convincing approach would be to purchase the top 10% of stocks based on cross-sectional prediction values, with equal weighting across them.

### Minor Issues:

1. Some of the terminology used in the paper is unconventional. For instance, "portfolio return" in Table 1 is more commonly referred to as "cumulative return," and "annual return" should be labeled as "annualized return." Additionally, the risk-free rate used in the analysis is not specified.

2. Several references on ensembled Q-learning have been published, but this paper lacks a rigorous discussion of these existing works.

**Questions:**

1. Please carefully reconsider how you structure your network inputs.
2. Please ensure that the comparison between the proposed method and classical methods is conducted fairly.

---

> ### Author Response · Authors · 2024-11-25
> **Responses to Reviewer jZAh (Part 1)**
>
> > 1. Lack of Discussion on Non-Stationarity.
>
> (1) "*As I understand it, the paper attempts to mitigate non-stationarity through specific data transformations.*"
>
> We would like to clarify that our approach mitigates the non-stationarity issue primarily from both the algorithmic and data perspectives, rather than relying solely on specific data transformations. The fundamental idea behind the bilevel optimization algorithm is to prevent overfitting during offline learning, ensuring that the model does not simply memorize the optimal policy based on specific patterns in the training set. Different data transformation methods are treated as plug-and-play modules within the bilevel learning framework, designed to generate meaningful augmented data that helps evaluate the robustness of the learned policies.
>
> Notably, the proposed bilevel optimization scheme was shown effective even without any specific data transformation methods (see Table 3). Below, we summarize the ablation study results:
>
> |  Bilevel learning |  Data transformation   | Cumulative return (CR) $\uparrow$     |  Annualized return (AR)  $\uparrow$  |  Sharpe ratio (SR) $\uparrow$  |
> | ----------- | ---- | ---- | ---- |---- |
> | No | No  | 1.46 | 0.35 | 1.37 |
> | Yes | No   | 1.66 | 0.385 | 1.44 |
> | Yes | Yes   | 1.84 | 0.417 | 1.61 |
>
> (2) "*The rationale behind the choice of these three particular transformations is unclear.*"
>
> The data transformation methods employed in our work are designed to simulate out-of-distribution yet plausible market changes that have not been included in the training set. From a time series analysis perspective, $F_{1,2,3}$ covers the short-term randomness, long-term trends, multi-scale seasonal patterns of a dynamic system, respectively. Please refer to our **General Response 1** for further details.
>
> (3) "*Applying this approach in the out-of-sample leads to the future information issue.*"
>
> We would like to clarify that there is no test data leakage in any of the data transformation methods used. For further details, please refer to our **General Response 1**.
>
> > 2. Input Construction and Covariance Matrix
>
> $o_{t-H+1:t}^\text{cov} \in \mathbb{R}^{|S|\times|S|}$ calculates the covariance matrix of returns between pairs of stocks over a fixed period of time $H$ before $t$. In practice, we adopted the settings used in StockFormer. For daily-level stock trading data, we set $H=1$ to capture short-term predictive features and cross-stock relational features, and $H=5$ to capture long-term predictive features. This matrix can partially capture the correlations between stocks and remains operational on larger datasets.
>
> Additionally, we have conducted experiments on another dataset containing more stocks (587 in total). The results indicate that our method is also scalable to larger datasets. Please see our **General Response 3** for details.

---

> > ### Author Response · Authors · 2024-11-25
> > **Responses to Reviewer jZAh (Part 2)**
> >
> > > 3. Experimental Shortcomings
> >
> > (1) "*To compute the true value theoretically, a globally optimal policy is required.*"
> >
> >
> > The so-called **true value** of a state or state-action pair is defined as the expected return when following a specific policy. The overestimation problem refers to the phenomenon where the estimated values are higher than the true values under the **current policy**, not the true values of an optimal policy. This discrepancy often arises due to data limitations and inherent biases in the value estimation process.
> >
> > (2) "*It would strengthen the results if the maximum drawdown (MDD) were also reported alongside the annual return.*"
> >
> > Thank you for your suggestion! We have included the following MDD results in the revised Appendix F.2. We summarize the results below (lower is better):
> >
> > |       | Market benchmark | HATR   | SARL          | FinRL-SAC      | StockFormer    | Metatrader     |
> > |--------------|------------------|--------|---------------|----------------|----------------|----------------|
> > | CSI-300      | 0.31            | 0.51   | 0.36±0.02     | 0.30±0.01      | 0.31±0.02      | 0.28±0.02      |
> > | NASDAQ-100   | 0.28            | 0.35   | 0.40±0.01     | 0.32±0.01      | 0.32±0.02      | 0.31±0.00      |
> >
> >
> > (3) "*The stock prediction strategy—buying only the single stock with the highest predicted value—is problematic, as it introduces excessive randomness into the strategy.*"
> >
> > Thank you for your suggestion. To provide a more comprehensive comparison, we test an additional strategy for stock prediction, namely uniformly buying the top 10% of stocks based on predicted returns. This helps reduce the randomness inherent in such methods to some extent. We follow the experimental setup from Table 1 and the specific results of CSI-300 dataset are presented below.
> > |                              | Cumulative return (CR) $\uparrow$ | Annualized return (AR)  $\uparrow$ | Sharpe ratio (SR) $\uparrow$ | Maximum drawdown (MDD) $\downarrow$ |
> > | ---------------------------- | --------------------------------- | ---------------------------------- | ---------------------------- | ----------------------------------- |
> > | HATR (Top 10%)               | -0.03                             | -0.01                              | 0.07                         | 0.49                                |
> > | Relational Ranking (Top 10%) | -0.16                             | -0.06                              | -0.06                        | 0.38                                |
> > | AutoFormer (Top 10%)         | -0.05                             | -0.02                              | 0.04                         | 0.55                                |
> > | FactorVAE (Top 10%)          | 0.89                              | 0.24                               | 1.11                         | 0.17                                |
> > | HATR (Top 1)             |  -0.05                                 |       -0.02                             |        0.06                      |     0.51                                |
> > |   Relational Ranking (Top 1)                           |       -0.13                            |    -0.05                                |   -0.05                           |           0.37                          |
> > |  AutoFormer (Top 1)                            | -0.08                                  |     -0.03                               |            0.02                  |              0.58                       |
> > |  FactorVAE (Top 1)                            | 0.96                                  |     0.25                              |      1.25                        |                                     0.17|
> > |  MetaTrader (Ours)                            |   1.44                                |     0.35                               |      1.35                        |        0.28                             |
> >
> >
> > > 4. Some of the terminology used in the paper is unconventional.
> >
> > We have revised the terminologies suggested by the reviewer in the revision. Thank you for the helpful suggestions! Additionally, in all experiments, the risk-free rate is set to 0 when calculating the Sharpe Ratio.
> >
> > > 5. Several references on ensembled Q-learning have been published, but this paper lacks a rigorous discussion of these existing works.
> >
> > We present the differences between our conservative TD method and other ensemble-based Q-learning methods in **General Response 2**, and have included this discussion in the **Appendix G** in the revision.

---

> > > ### Comment · Reviewer_jZAh · 2024-11-26
> > > **Further Feedback**
> > >
> > > Thank you to the authors for your thoughtful responses. After reviewing the comments from other reviewers and considering your replies, I believe there are still some unresolved issues in the revised manuscript.
> > >
> > > 1. I am still unclear about the design of the data transformation method $F_1$. Specifically, why does $F_1$ only consider the top 10% of stocks and not the bottom 10% as well? The authors argued that $F_1$ aims to answer a key question: How would MetaTrader respond if the top 10% of the most profitable stocks suddenly collapsed the following day? In finance, this phenomenon is called short-term reversal, and it is symmetric. The bottom 10% of stocks also have the potential for a strong surge, which seems to be overlooked.
> > >
> > > 2. Based on the authors' response, it appears that the sample covariance matrices are calculated from either one-day or five-day observations. For instance, in the CSI-300 dataset, you use at most $300 \times 5$ data points to estimate a $300 \times 300$ matrix. Needless to say, this leads to biased covariance matrix estimates. In my opinion, an ablation study examining the impact of using the covariance matrix as an input would be necessary to address this potential bias.
> > >
> > > 3. Regarding minor suggestions, comparing the proposed algorithm with ensemble Q-learning methods may enhance the quality of this paper.
> > >
> > > I hope these comments are helpful in improving your paper.

---

> ### Author Response · Authors · 2024-11-27
> **Further Reponse to Reviewer jZAh**
>
> We would like to thank the reviewer for the prompt reply and ongoing discussion!
>
> > 1. Further explanation on transformation method $F_1$.
>
> First, following previous RL-based stock trading methods, our trading operations only include buying, holding, or selling, without considering short-selling. Thank you for your valuable suggestion regarding the action space allowing for both long and short positions. We believe this is a significant point, and it will be discussed in future work.
>
> In our work, the data transformation method $F_1$ is primarily designed to prevent overfitting to the in-domain optimal but actually impractical policies learned exclusively on training data. As shown in Table 3 in the manuscript, compared to "$\textit{w/o}$ Data Transformation", the sharpe ratio (SR) improves significantly with the $F_1$ (1.44 $\rightarrow$ 1.53). Additionally, we further test the maximum drawdown (MDD), which decreases by $16 \\%$ (0.25 $\rightarrow$ 0.21), indicating that this approach can mitigate the trading risk.
>
> We were unsure whether inverting the bottom $10\\%$ would yields the same improvement in policy robustness, as short-selling was not included in the action space. But still, following the reviewer's suggestion, we add an experiment that inverts both the top and bottom $10\\%$ of stock returns. We present the comparison results under the online adaptation setup on the CSI-300 dataset below. The results can be fairly compared with those in Table 3 in the manuscript.
>
> | | Cumulative return (CR) $\uparrow$ | Annualized return (AR)  $\uparrow$ | Sharpe ratio (SR) $\uparrow$ | Maximum drawdown (MDD) $\downarrow$ |
> | -| - | - | - | -- |
> | w/o Data Transformation | 1.66 | 0.385 | 1.44 | 0.25 |
> | Top $10\%$ ($F_1$)      | **1.69**  | **0.390**| **1.53**| **0.21**   |
> | Bottom $10\%$ & Top $10\%$  | 1.67  | 0.387   | 1.47  | 0.24|
>
>
> > 2. Clarification on covariance matrix.
>
> We sincerely apologize for the previous explanation regarding the covariance matrix, which may have caused the misunderstanding that only a short historical period was used for its calculation. In our previous response, the fixed period $H$ refers to the **future time interval** used to calculate stock price changes, not the number of lookback days.
>
> In fact, we use the "price change matrix" (i.e., the daily price changes of all stocks over 252 historical trading days) to compute the covariance matrix. Below, we present the formulas used in the calculation process.
>
> Predefined parameters:
> - Lookback length $T$ (252 days in practice),
> - Fixed period length $H$,
>     - $H=1$ to capture short-term predictive features and cross-stock relational features,
>     - $H=5$ to capture long-term predictive features
> - Number of stocks $|S|$.
>
> The price change matrix $R_{tj}\in \mathbb{R}^{T\times|S|}$, for $r_{tj}\in R_{tj}$,
> $$
> r_{tj} = \frac{P_{t,j} - P_{t-H,j}}{P_{t-H,j}}
> $$
> The covariance matrix $\Sigma\in\mathbb{R}^{|S|\times|S|}$, for $\Sigma_{ij}\in \Sigma$
> $$
> \Sigma_{ij} = \frac{1}{T-1} \sum_{t=1}^{T} \left( R\_{ti} - \bar{R}\_i \right) \left( R\_{tj} - \bar{R}\_j \right)
> $$
>
> On the other hand, as you point out, the lookback length is likely to impact the calculation of the covariance matrix, which in turn can affect the final results. To further analyze the impact of lookback length, we conduct additional experiments and present the offline evaluation results on the CSI-300 dataset as follows:
>
> | Lookback length| Cumulative return (CR) $\uparrow$ | Annualized return (AR)  $\uparrow$ | Sharpe ratio (SR) $\uparrow$ | Maximum drawdown (MDD) $\downarrow$ |
> | -| -| - | -| - |
> |  20  | 0.81| 0.22 | 0.88| 0.38|
> | 126 | 1.39 | 0.34| 1.34| 0.29  |
> |   252 (Ours)  | **1.44** |  **0.35** |  **1.35** | **0.28**|
>
> As we can see from the above results, an insufficient lookback length significantly affects the estimation of the covariance matrix, ultimately leading to a noticeable decline in performance. Additionally, the lookback length we selected allows for a more accurate estimation of the covariance matrix, thereby ensuring the stability of the model’s performance.
>
> > 3. Regarding minor suggestions, comparing the proposed algorithm with ensemble Q-learning methods may enhance the quality of this paper.
>
> Thank you for your suggestions. We conduct new experiments by replacing the ensemble method in our model with the methods [1,2] compared in **General Response 2**, i.e., using the minimum and mean value of five parallel Q-networks as the Bellman target. The results are presented below, and are included in **Appendix G** in the revision.
>
>
> | Ensemble method      | Cumulative return (CR) $\uparrow$ | Annualized return (AR)  $\uparrow$ | Sharpe ratio (SR) $\uparrow$ | Maximum drawdown (MDD) $\downarrow$ |
> | - | - | -| -| -|
> |  Minimum value [1] | 1.17| 0.30  | 1.03 | 0.34|
> | Mean value [2] | 1.10| 0.28  | 0.99| 0.33 |
> |  Ours|    **1.44**  |        **0.35**  |                   **1.35**           |    **0.28**                                 |

---

### Official Review · Reviewer_Q9cS · 2024-10-21

**Soundness:** 3
**Presentation:** 3
**Contribution:** 3
**Rating:** 8
**Confidence:** 4

**Summary:**

This paper studies a way to train an RL policy for stock trading using past transaction data. To deal with the non-stationary and out-of-distribution data in the deployment phase, the paper proposes to perform (i) data transformation (augmentation), (ii) updating the Q function via pessimistic ensemble among multiple subsets of (augmented) data, and (iii) fine-tuning the data collected in the most recent time period. The experiment results demonstrate that the proposed method works better than existing online and offline RL approaches in the real stock transaction data.

**Strengths:**

- **Data transformation (augmentation) makes most of the application-specific features of finance.**

As a key component of the proposed approach, the paper proposed to perform data augmentation to enhance the generalizability of the model to out-of-distribution (OOD) data. In many RL settings, this is usually not a promising approach due to the difficulty of predicting/obtaining rewards for the OOD (state, action) pairs.

However, in the finance application, this is not the case because we can assume individual actions do not make a huge impact on the market dynamics, and what depends on the actions (selling or buying) is only the individual reward they get from these transactions. This paper nicely exploits this structure and does data augmentation.

- **Approaches, including bi-level optimization and fine-tuning, sounds reasonable and the components are harmoniously combined.**

The paper combines the aforementioned components in a sound way, and there are no components that seem redundant or unnecessary. The motivations behind why the proposed method needs each component are also well explained. While each component itself (e.g., fine-tuning and conservative learning) is not very novel, yet, I think this idea is worth sharing.

- **Well done ablations and experiments.**

The baselines are picked from representative algorithms of approaches in predictions, online and offline RL. Also, ablations on w/ and w/o data transformation, w/ and w/o fine-tuning, and runtime comparison are all informative for readers.

- **Clarity of the paper.**

Overall, the paper is clearly written and easy to follow.

**Weaknesses:**

To be honest, there is not much to point out for weakness. However, the following might have room for improvement.

- **Related work on offline RL and offline-to-online RL**

The current related work focuses on the existing approaches for learning a stock-trading policy. Adding discussion on why the conventional offline RL does not work and why the proposed method works might be helpful for readers who are not very familiar with offline RL. Also, there is some work on offline-to-online RL, which does online fine-tuning after performing offline RL. This approach is relevant to the proposed method, and can be worth mentioning in the related work section.

- **Description about the proposed method in Introduction**

While the paper was in general very well written, the description of the proposed method in the introduction may have some room for improvement. It was difficult for me to imagine what the proposed method looks like only from the introduction (in particular, it was not clear to me what "in-domain profits" refer to and what the outer-loop optimization does), while It became clear after reading Section 3.

**Questions:**

- How does the proposed method (and other baseline) determine the trajectory length? Are there any guidelines to pick a reasonable trajectory length if there are some tradeoffs between short and long trajectories?

---

> ### Author Response · Authors · 2024-11-25
> **Responses to Reviewer Q9cs**
>
> > 1. Related work on offline RL and offline-to-online RL
>
> Thank you for your encouraging comments and valuable suggestions! We have included a further discussion on the related work in Section 5 in the revision.
>
>
> > 2. Description about the proposed method in Introduction.
>
> Thanks again for your suggestion! We have updated the descriptions of "in-domain profits" and "outer-loop optimization" in the revised Introduction section.
>
>
> > 3. How does the proposed method (and other baseline) determine the trajectory length?
>
> For the baseline models, we directly used their official implementations and adopted the trajectory length settings as specified in their original literature. For our approach, we followed the same trajectory length as used in StockFormer.
>
> Specifically, during the training phase, the trajectory has two termination conditions: reaching a maximum predefined length or the last day of the training set. In the original manuscript, we set the same configuration as StockFormer, defining a maximum length threshold of 1000 with a random start time. Additionally, we conduct experiments to analyze the impact of using different trajectory lengths threshold in our model (corresponding to the experimental setup in Table 1).
>
> | Maximum trajectory length | Model                   | CR       | AR       | SR       |
> | ------------------------- | ----------------------- | -------- | -------- | -------- |
> | 100                       | MetaTrader (ours)            | 1.34     | 0.33     | 1.23     |
> | 500                       | MetaTrader (ours)             | 1.36     | 0.33     | 1.26     |
> | 1000                      | MetaTrader (ours)             | **1.44** | **0.35** | **1.35** |
> | 1000                      | StockFormer (prior art) | 1.24     | 0.31     | 1.20     |

---

> > ### Comment · Reviewer_Q9cS · 2024-12-01
> >
> > Thank you for the response. I have read it.

---

### Official Review · Reviewer_hygz · 2024-11-03

**Soundness:** 2
**Presentation:** 3
**Contribution:** 2
**Rating:** 3
**Confidence:** 4

**Summary:**

This paper applies RL to portfolio management, focusing on generalization in dynamic financial markets. Traditional RL models often overfit offline data, leading to rigid policies. To address this, the authors propose MetaTrader, introducing a bilevel actor-critic approach and a conservative TD learning variant with an ensemble-based target to reduce value overestimation. Experiments on two public datasets indicate that MetaTrader achieves improvements over existing RL and stock prediction models, suggesting enhanced adaptability in changing markets.

**Strengths:**

•	OOD (out-of-distribution) issues are prevalent in the stock market. This paper focuses on augmenting offline data to train a policy with stronger generalization capabilities to handle distribution shifts in online stock data. This is a crucial and interesting topic.

•	The writing is logically structured and easy to follow, making the content clear and accessible to readers. The arguments are presented in a straightforward manner, allowing readers to grasp the key points without difficulty.

•	The paper includes appropriate experiments and ablation studies to support its findings. These experiments provide evidence for the proposed method’s effectiveness, though certain aspects of the experimental setup could be further improved for a more comprehensive evaluation.

**Weaknesses:**

•	Relying on training the policy with real and augmented data, followed by fine-tuning on real data to address the OOD problem in the stock market, is not an ideal approach. This can be observed in the following two aspects:

1) The augmentation methods have limitations. For instance, do the F2 and F3 methods mentioned in the paper potentially introduce data leakage? Additionally, is the choice of the top 10% in F1 (why specifically 10%?) reasonable?

2) The effectiveness of fine-tuning is questionable. Based on findings in the application of RL to finance, the performance of fine-tuning RL can be similar to or even weaker than training RL from scratch. The experiments in the paper do not include a comparison between fine-tuning RL and training RL from scratch to validate its effectiveness.

Note: To clarify, “training from scratch” here refers to directly using the original data for portfolio management without involving data transformations in training.

•	The experimental setup has some flaws, detailed as follows:

1) Data Validity: The experimental data only goes up to 2022, without including more recent data (e.g., up to 2024), which makes the findings less convincing. Websites like Yahoo Finance (https://finnhub.io/), Alpaca (https://docs.alpaca.markets/), and FMP (https://site.financialmodelingprep.com/developer/docs#chart-intraday) provide easy access to the latest data, which would strengthen the study’s confidence.

2) Stock Selection and Market Representation: While the CSI300 index theoretically consists of 300 stocks, only 88 were selected in this study. Additionally, both chosen stock markets consist of only around 80 stocks, without distinguishing between different market scales. This limited selection makes the results less persuasive, and it would be beneficial to include stock markets of varying scales for a more comprehensive analysis.

3) Evaluation Metrics: The choice of metrics is somewhat limited. The study only focuses on return-related metrics (even though the Sharpe Ratio accounts for risk-adjusted returns, it still primarily measures profitability). Including risk metrics such as VOL and MDD would provide a more balanced evaluation of performance.

Note: I noticed that this paper follows the experimental setup of StockFormer. However, using data from two years ago is not ideal and updating the data could significantly improve the credibility of the experiments. Furthermore, StockFormer included MDD as a risk metric, so it’s unclear why this study chose not to incorporate it in its evaluation criteria.

•	Benchmark Selection**.** Many financial companies still use machine learning and deep learning methods based on prediction, such as LightGBM, LSTM, and Transformer models. Including these in the experiments would provide a more convincing comparison. Qlib (https://github.com/microsoft/qlib) can be a helpful reference for implementing these methods.

**Questions:**

•	As described in the weaknesses section, is there a potential issue of data leakage with the three types of data transformations mentioned? Additionally, why is it set to the top 10%? It would be helpful if the authors could clarify these points.

•	In Figure 3, the augmented data is labeled from 1 to M, while the finetuning data is labeled as M+1 to M'. If I understand correctly, should it actually be M+1 to M+M'? If this is not a typo, please disregard this comment. If this is indeed a writing error, it would be helpful for the authors to adjust any other relevant parts of the paper accordingly.

•	Lines 87–94 mention that the state data consists of action-free and action-dependent components. I would like to know more about the data processing methods used. For example, variables like price, cash holdings, and position can exhibit significant differences over extended trading periods. For instance, Apple’s (AAPL) price on the NASDAQ was around $12 in 2011, but by 2022, it had exceeded $130. This shift in stock prices can lead to substantial differences in cash, position, and price scales. Without normalization, how is RL training stability maintained? It would be helpful if the authors could explain this in detail.

---

> ### Author Response · Authors · 2024-11-25
> **Responses to Reviewer hygz (Part 1)**
>
> > 1. Relying on training the policy with real and augmented data, followed by fine-tuning on real data to address the OOD problem in the stock market, is not an ideal approach.
>
> **(1) Do the F2 and F3 methods mentioned in the paper potentially introduce data leakage? Is the choice of the top 10% in F1 (why specifically 10%?) reasonable?**
>
> Please refer to our **General Response 1**.
>
> **(2) The effectiveness of finetuning of RL-for-finance models.**
>
> Thank you for the insightful comment! We completely agree with the reviewer that in practical RL-for-finance tasks, the naive fine-tuning approach often fails to enhance model performance on test data. This is primarily due to overfitting to specific data patterns when fine-tuning on more recent data. This is precisely why we propose the bilevel optimization approach for the RL method!
>
> Theoretically, the bilevel optimization scheme can significantly enhance the model's generalizability to new data. Similar approaches, known as model-agnostic meta-learning (MAML) [1], have been widely adopted to improve finetuning results in few-shot learning scenarios. Intuitvely, it aims to find optimal parameters initialization that can be quickly adapted to a new related task using only a few data and a few gradient steps.
>
> **(3) Finetuning vs. Train-from-scratch.**
>
> We compare the performance of different RL methods with and without finetuning, using the same configuration as Table 1. Below, we present the Cumulative Return (i.e., PR in the original manuscript) on the CSI-300 dataset. The results are averaged over three random training seeds. Notably, our bilevel optimization approach significantly improves the finetuning results (by **+13.39%**), while the previous RL approaches do not support such effective model finetuning (e.g., by +0.81% for StockFormer). We include full comparisons in the revised appendix.
>
>
> |                      | Finetune | Train from scratch  | Promotion by finetuning|
> | -------------------- | --------------------- | ---- |-|
> | SARL                 | 1.06 $\pm$ 0.14          | 1.03 $\pm$ 0.13  | +2.91%|
> | CQL                  | 0.64 $\pm$ 0.07    | 0.69 $\pm$ 0.05  | -7.25%|
> | IQL                  | 1.02 $\pm$ 0.10         |0.96 $\pm$ 0.10  | +6.25%|
> | FinRL-SAC            | 0.83 $\pm$ 0.05        | 0.80 $\pm$ 0.07  |+3.75%|
> | FinRL-DDPG           | 0.58 $\pm$ 0.15    | 0.63 $\pm$ 0.13  |-7.94%|
> | StockFormer          | 1.24 $\pm$ 0.10     | 1.23 $\pm$ 0.09  |+0.81%|
> | MetaTrader           | **1.44 $\pm$ 0.07**     | **1.27 $\pm$ 0.08**  |**+13.39%**|
>
> Reference:
> [1] Finn C., Abbeel P., Levine S. Model-agnostic meta-learning for fast adaptation of deep networks. ICML 2017.
>
>
> > 2. On the experimental setup.
>
> **(1) Data validity: The experimental data only goes up to 2022, without including more recent data (e.g., up to 2024).**
>
> In our original experiments, we used data up to 2022 to ensure a fair comparison with StockFormer, which follows the same training and testing period division.
>
> As suggested by the reviewer, we conduct additional experiments using data beyond 2022. In this experiment, we do not extend the training set range but directly test on the CSI-300 dataset spanning from 2022-05-01 to 2024-05-01. During this period, the overall market is weaker than that in the original test set before 2022. Consequently, the annualized returns of all methods are reduced. Nonetheless, our method consistently outperforms all baselines, highlighting its potential for profitability even under more challenging market conditions. We have included these results in Appendix F.3 in the revision.
>
> |             | Cumulative return (CR) $\uparrow$ | Annualized return (AR)  $\uparrow$ | Sharpe ratio (SR) $\uparrow$ | Maximum drawdown (MDD) $\downarrow$ |
> | ----------- | ------- | ------- | ---- | ---- |
> | SARL        | -0.13     | -0.07           | -0.07    | 0.51   |
> | FinRL-SAC   | 0.04     | 0.01      | 0.03    | 0.49                                |
> | StockFormer | 0.21                             | 0.10                               | 0.46                         | 0.45                                |
> | MetaTrader  | **0.32**                             | **0.15**                               | **0.76**                         | **0.44**                                |
>
>
> **(2) Stock selection and market representation.**
>
> Please refer to our **General Response 3**.
>
> **(3) Evaluation metrics.**
>
> Thank you for your suggestion! We have included the following MDD results in the revised Appendix F.2. We summarize the results below (lower is better):
> |       | Market benchmark | HATR   | SARL          | FinRL-SAC      | StockFormer    | Metatrader     |
> |-----|--|----|------|------|---|------|
> | CSI-300      | 0.31            | 0.51   | 0.36±0.02     | 0.30±0.01      | 0.31±0.02      | 0.28±0.02      |
> | NASDAQ-100   | 0.28            | 0.35   | 0.40±0.01     | 0.32±0.01      | 0.32±0.02      | 0.31±0.00      |

---

> > ### Author Response · Authors · 2024-11-25
> > **Responses to Reviewer hygz (Part 2)**
> >
> > > 3. Benchmark selection: Many financial companies still use machine learning and deep learning methods based on prediction. Including these in the experiments would provide a more convincing comparison.
> >
> > Thank you for the suggestion. First, we would like to kindly remind the reviewer that our original manuscript already includes the following prediction-based benchmarks in the experiments:
> > - HATR (2021)
> > - Relational Ranking (2019)
> > - AutoFormer (2021), which is a Transformer-based model
> > - FactorVAE (2022), which is specifically designed for stock prediction
> >
> > Among these methods, FactorVAE achieves the best performance but still significantly underperforms our approach (Cumulative Returns: 0.96 vs. 1.44 in Table 1; 1.07 vs. 1.84 in Table 2).
> >
> > Additionally, we have now included LightGBM, LSTM, and Transformer for comparison in the updated experiments:
> >
> > |             | Cumulative return (CR) $\uparrow$ | Annualized return (AR)  $\uparrow$ | Sharpe ratio (SR) $\uparrow$ | Maximum drawdown (MDD) $\downarrow$ |
> > | ----------- | -------------------------------- | ---------------------------------- | ---------------------------- | ----------------------------------- |
> > | LightGBM    | -0.14                            | -0.05                              | 0.03                         | 0.49                                |
> > | LSTM        | 0.09                             | 0.02                               | 0.19                         | 0.51                                |
> > | Transformer | 0.19                             | 0.06                               | 0.34                         | 0.41                                |
> > |   MetaTrader         |    **1.44**                              |        **0.35**                            |                   **1.35**           |    **0.28**                                 |
> >
> >
> > > 4. In Figure 3, the augmented data is labeled from 1 to M, while the finetuning data is labeled as M+1 to M'.
> >
> > Thank you for pointing this out. We have corrected this in the revision.
> >
> > > 5. Without normalization, how is RL training stability maintained? It would be helpful if the authors could explain this in detail.
> >
> > In practice, normalization is required. We perform normalization separately for each stock, ensuring that all normalization factors are specific to the data of the individual stock. Any stock with complete data can be normalized individually and the normalization of each stock will not affect the others. For a given stock, all price data (open, close, high, low) share the same normalization factor. We have added the details of the normalization process in the revision of Appendix B.3.

---

> > > ### Comment · Reviewer_hygz · 2024-11-26
> > >
> > > Thank you for your detailed response. After thoroughly reviewing the comments from other reviewers and reflecting on your replies, I deeply appreciate the additional experiments you conducted to address the concerns and provide further clarification. However, I still have some remaining questions and concerns:
> > >
> > > 1. I still have some concerns regarding the use of data augmentation and finetuning to address OOD issues, specifically about how data augmentation can ensure consistency with real market dynamics.
> > > 2. Regarding the data scale, the size difference between the two stock pools selected is relatively small. Considering the limited time for discussion at this stage, I hope the authors can provide comparative results on stock pools of different scales in the future.
> > > 3. Regarding data normalization, the authors provided the normalization method for action-free data. However, I am more interested in understanding how normalization is performed for action-dependent data. A detailed explanation would be greatly appreciated if possible.

---

> > > > ### Author Response · Authors · 2024-11-27
> > > > **Further Reponse to Reviewer hygz**
> > > >
> > > > We thank the reviewer for the prompt feedbacks.
> > > >
> > > > > 1. Further discussion on data transformation
> > > >
> > > > We have already explained in the **General Response** that the data transformation method was designed to account for possible variations in the real market. The experimental results in Table 3 also demonstrate that $F_{1,2,3}$ are indeed effective in practice.
> > > >
> > > > We are unsure why the reviewer places significant emphasis on the importance of keeping the augmented data perfectly aligned with real market dynamics. In our view, **maintaining perfect consistency is unnecessary** for two key reasons:
> > > > - We do perform meta-finetuning on **real data** (referred to as "*in-distribution model finetuning*"). Thanks to the prior bilevel policy learning stage, our model parameters are able to quickly adapt to real data. As a result, it is unnecessary for the augmented data in the bilevel learning stage to perfectly match the real data distribution. This "bilevel learning + meta-finetuning" approach has been extensively validated as more effective than conventional finetuning methods, especially in few-shot or domain adaptation scenarios. We encourage the reviewer to refer to the prior literature on bilevel gradient-based meta-learning approaches for further context [1,2].
> > > > - Not all effective augmentation methods need to strictly preserve complete consistency between the transformed and original data. For instance, widely used augmentation techniques in image recognition, such as **color adjustment** and **image flipping**, are highly effective in practice. Yet, are they entirely consistent with the original training data distribution? Certainly not! The purpose of our data augmentation methods is to enhance the robustness of the policy in fictitious worst-case scenarios. While this may introduce some errors, it represents a variance-bias trade-off: training the agent on a fixed, small dataset would lead to significant variance in the value function due to overfitting. In our case, addressing overfitting is clearly the higher priority.
> > > >
> > > > References:
> > > >
> > > > [1] Finn et al. Model-agnostic meta-learning for fast adaptation of deep networks. In ICML, 2017.
> > > >
> > > > [2] Mitchell et al. Offline metareinforcement learning with advantage weighting. In ICML, 2021.
> > > >
> > > >
> > > > > 2. Dicussion on a larger market scale.
> > > >
> > > > Please refer to our **General Response 3**.
> > > >
> > > > > 3. Regarding data normalization, the authors provided the normalization method for action-free data. However, I am more interested in understanding how normalization is performed for action-dependent data.
> > > >
> > > > For the action-free data, we normalize the original data, including **open price, close price, high price, low price, and trading volume**, and all technical indicators are computed based on the normalized values.
> > > >
> > > > The normalization method for action-dependent data, however, follows the methods used in previous work including FinRL and StockFormer.
> > > >
> > > > The action-dependent state space is processed independently from the raw data, such as prices and volume, and is derived solely from the action space. Specifically, the action space vector represents the buy, sell, or hold actions for each stock. Each element of the vector corresponds to the action proportion for the respective stock, typically within the range of $[-1, 1]$. Positive values represent buying, negative values represent selling, and a value of zero indicates holding the current position. In practice, all positive values are normalized to represent the proportion of remaining funds used to purchase each stock, while all negative values are kept within $[-1, 0)$, representing the proportion of stock holdings to be sold. We calculate the total account balance using the actual prices and determine the stock holdings using the normalized action vector. These two components are then concatenated to form $z_t\in\mathbb{R}^{1+|S|}$ in the state space.

---

### Official Review · Reviewer_MUZC · 2024-11-03

**Soundness:** 2
**Presentation:** 2
**Contribution:** 2
**Rating:** 3
**Confidence:** 3

**Summary:**

The paper introduces "MetaTrader," a reinforcement learning (RL) approach for stock trading that aims to overcome challenges of overfitting to offline data and lacking generalizability in non-stationary financial markets. MetaTrader trains policies to perform well on transformed out-of-distribution (OOD) data, with a conservative temporal difference (TD) ensemble, designed to mitigate value overestimation in offline RL. Through bilevel actor-critic training on both original and transformed stock data, MetaTrader builds robustness to OOD scenarios. Empirical results on two stock datasets indicate MetaTrader's superiority over existing RL and stock prediction methods, with improved performance in portfolio returns and risk-adjusted metrics like Sharpe ratio, demonstrating its potential for robust and adaptable trading strategies.

**Strengths:**

This paper addresses an interesting and important problem: stock trading in the offline RL setting. The bilevel optimization approach combined with conservative TD learning directly addresses practical challenges in offline RL for trading, particularly the out-of-distribution (OOD) generalization problem.

The paper effectively positions stock trading as a unique offline RL problem, underscoring the potential for dataset expansion through their decoupled MDP framework.

 The empirical results demonstrate a comparison with other existing models, including SARL, FinRL-SAC, FinRL-DDPG, and StockFormer. Real-world data sets, specifically the CSI-300 Index and NASDAQ-100, are used.

**Weaknesses:**

The notations are not clearly explained and are sometimes used inconsistently, which makes them hard to follow. Some points related to clarity are outlined below.

1. Observation space on page 2: “$K$ technical indicators that reflect the temporal trends of stock prices”. There is no explanation of what technical indicators are or how they reflect the temporal trends.

2. State space on page 2: “The action-free market state $h_t$ is composed three types of latent states $s_t^{\text relat}$, $s_t^{\text long}$, $s_t^{\text short}$ generated from the observation $o_t^{\text price}$, $o_t^{\text stat}$, $o_t^{\text conv}$”. There is no explanation about how $s_t$ is generated by $o_t$. Instead, the authors refer to Eq (1) on page 3, however, Eq (1) does not include $s_t$.

3. Dimension of $h_t$ on page 3: The dimension of $h_t$ is stated as the number of stocks times $D$, but there is no explanation of what the notation $D$ represents.

4. Daily prices in Appendix A: The input data for daily prices are denoted by $p_t$. This should be written as $o_t^{\text close}$. The notations are used inconsistently.

5. Metrics in Appendix C: The evaluation metrics for the experiments are defined in Equations (4), (5), and (6). The definitions are very vague and the notations are not clearly explained, lacking connection to $o_t$, $h_t$, $z_t$, $a_t$ defined in Section 2.


The authors manipulate the input data using the three transformation methods. While the paper notes that data transformations are used to create diverse subsets, it does not provide a clear rationale for the chosen transformations. These transformations can substantially impact the underlying factors driving stock prices and alter correlations between stocks, so an explanation of their selection and intended effects would strengthen the work. Appendix A explains their methods; however, there is no justification or theoretical background provided. Although the empirical studies demonstrate that the proposed algorithm outperforms the existing methods using two real-world data sets, the benefits of using the proposed algorithm are not clear to me, because it functions like a black-box solution.


Using an ensemble-based conservative TD target to mitigate overestimation in offline RL, though effective, lacks novelty, as this technique has been explored extensively in previous research.


Since this work primarily emphasizes empirical results over theoretical contributions, the experimental section should be more robust. For instance, the significance of the findings in Tables 3, 4, and 5 cannot be interpreted without standard deviations. To provide a clearer picture of the proposed algorithm’s stability, the authors could use a rolling training/test set and report both the mean and standard deviation of performance metrics. This would offer a more comprehensive view of the algorithm’s consistency and reliability.

**Questions:**

Please see the weaknesses listed above.

---

> ### Author Response · Authors · 2024-11-25
> **Responses to Reviewer MUZC**
>
> > 1. On clarity of the notations.
>
> **(1) Details about technical indicators**
>
> The technical indicators mentioned in the paper follow the settings used in StockFormer. Specifically, we utilized the Stockstats package, and the technical indicators are presented below. We include this in the Appendix E in the revised paper.
>
> | Technical Indicator | Description                                        |
> |----------------------|-----------------------------------------------------|
> | macd                 | Moving average convergence divergence               |
> | boll_ub              | Bollinger bands ( upper band )                        |
> | boll_lb              | Bollinger bands ( lower band )                        |
> | rsi_30               | 30 periods relative strength index                  |
> | cci_30               | Retrieves the 30 periods commodity channel index    |
> | dx_30                | Directional index with a window length of 30        |
> | close_30_sma         | 30 periods simple moving average of the close price |
> | close_60_sma         | 60 periods simple moving average of the close price |
>
>
>
>
> **(2) State space on Page 2**
>
> Thank you for pointing this out. The three types of latent states should be represented as $h_t^\text{relat}$, $h_t^\text{long}$, $h_t^\text{short}$. We have corrected this in the revision.
>
> **(3) Dimension of $h_t$ on Page 3**
>
> $D$ represents the dimension of the output by the feature extraction module.
>
> **(4) Daily prices in Appendix A**
>
> As suggested by the reviewer, we have corrected the notation for "closing price" to $o_t^{close}$ in the revision to align with the main text.
>
> **(5) Metrics in Appendix C**
>
> Thank you for your suggetion. In the revision, we have updated equations (7), (8), and (9) in Appendix C. We have provided a more detailed explanation of these notations and clarified the calculation of each metric.
>
> > 2. Rationale for the chosen transformations.
>
> Please refer to our **General Response 1**.
>
> > 3. Novelty of ensemble-based conservative TD target.
>
> Please refer to our **General Response 2**.
>
> > 4. To provide a clearer picture of the proposed algorithm’s stability.
>
> **(1) Standard deviations**
>
> As suggested by the reviewer, we have added the standard deviations for the experiments reported in Tables 3, 4, and 5.
>
> **(2) Rolling training/test set**
>
> In Lines 360--368 in the original manuscript, we have evaluated MetaTrader under the online adaptation setup, where all compared models are continuously finetuned on the streaming test data. Implemenatation details are provided in Appendix B. We understand this as using a rolling approach for model updates and testing. If we have misunderstood the reviewer's meaning of "rolling training/test," we kindly request more specific experimental suggestions.
>
> **(3) Maximum drawdown**
>
> To further showcase the stability of our approach, we have included the MDD (Maximum Drawdown) metric in Table 11 in the revised appendix, as an indicator of downside risk over a specified time period. We summarize the results below:
> | Dataset      | Market benchmark | HATR   | SARL          | FinRL-SAC      | StockFormer    | Metatrader     |
> |--------------|------------------|--------|---------------|----------------|----------------|----------------|
> | CSI-300      | 0.31            | 0.51   | 0.36±0.02     | 0.30±0.01      | 0.31±0.02      | 0.28±0.02      |
> | NASDAQ-100   | 0.28            | 0.35   | 0.40±0.01     | 0.32±0.01      | 0.32±0.02      | 0.31±0.00      |

---

### Author Response · Authors · 2024-11-25
**Revision Uploaded**

We thank all reviewers for the constructive comments and have updated our paper accordingly. Please check out the new version!

Important changes and new results include:
1. Clarify the technical novelty of this work (Section 1).
2. Explain the key inspirations behind the proposed data transformation methods (Section 3.2).
3. Compare our method with existing ensemble-based Q-learning methods (Section 3.3 and Appendix G).
4. Add the standard deviations of experiments with multiple random seeds (Tables 3, 4, and 5).
5. Add discussions and experiments on a larger market scale (Section 4.4 and Appendix F.1).
6. Add risk evaluation by Maximum Drawdown (Appendix F.2).
7. Add evaluation on more recent data (Appendix F.3).
8. Analyze the effectiveness of model finetuning for the RL-for-Finance methods (Appendix F.4).

Minor changes:
1. Add a footnote to provide a detailed explanation of the meaning of "in-domain profit" in Section 1.
2. Correct the notation of the latent states in Section 2.
3. Add a description of notation $D$ in Section 3.1.
4. Modify the superscript of the data range $M+M^{\prime}$ in Figure 3.
5. Modify terminologies in Section 4: Portfolio Return -> Cumulative Return, Annual Return -> Annualized Return.
6. Include prior work for offline-to-online RL in Section 5.
7. Modify the notations in Appendix A to align with the notations in the main text.
8. Clarify the data normalization method used in our model (Appendix B.3).
9. Polish the calculation formula for the evaluation metrics (Appendix C).
10. Introduce details of the technical indicators (Appendix E).

We appreciate the great efforts by the AC and reviewers. We are very confident that our paper is qualified for ICLR in both novelty and experimental results. Please do not hesitate to let us know for any additional comments on the paper.

---

### Author Response · Authors · 2024-11-25
**General Response (Part 1)**

> 1. To Reviewer MUZC, hygz, jZAh: Why are these data transformation methods chosen?

**(1) Why do we use data transformation? How do we handle non-stationarity?**

One primary contribution of our work is to enhance the generalization capability of the policy. In practice, we approach this from both the algorithmic and data perspectives, which are closely intertwined.
- From the algorithmic perspective, we adopt the bilevel optimization algorithm to prevent overfitting during offline learning, in the sense that keeping the model from simply memorizing the optimal policy based on specific patterns in the training set.
- From data perspective, our data transformation methods fundamentally aim to generate meaningful augmented data that aligns well with the bilevel optimization algorithm.

Specificially, we employ data transformations to simulate the extreme changes of the highly complex and stochastic stock market dynamics.

**(2) Why are these transformation methods chosen?**

We acknowledge that an unsuitable data transformation method may introduce extra noise and negatively impact the model's performance. In our approach, we treat stock market data as multivariate time series, those dynamic patterns can be typically viewed as a combination of three components:
- Short-term randomness,
- Long-term trends,
- Multi-scale seasonal patterns.

Accordingly, the three data transformation methods employed in our work are designed to simulate out-of-distribution (OOD) yet plausible market changes that have not been included in the training set, with each method focusing on one of the three dynamic components:
- $F_1$ **for OOD short-term randomness**: This method attempts to answer a key question: *How would MetaTrader respond if the top 10% most profitable stocks (a common threshold in stock trading to balance profit and risk) suddenly collapsed the following day? Can our policy effectively adapt to such a scenario?* $F_1$ is specifically designed for this purpose. It modifies the prices of the top 10% stocks, simulating the effects of unexpected events on individual stocks. Based on this, the bilevel learning scheme prevents the policy from overfitting to the stocks that perform exceptionally well during the training periods.
- $F_2$ **for OOD long-term trends**: This transformation is driven by such a question: How would our policy perform when faced with changes in the overall temporal trends of the stocks? By simulating varying market conditions influenced by long-term disruptions, it evaluates the policy's robustness in navigating such scenarios.
- $F_3$ **for OOD multi-scale correlations**: This transformation scales the seasonal patterns of the market by downsampling the training data, enabling the model to capture multi-scale correlations between the stock changes.

**NOTE: No test data leakage occurs with these transformation methods!** We apologize for the misleading description of $F_3$ in our original submission (Line 185): "*We downsample the original time series by four and concatenate the squeezed sequences with subsequent data.*" This was an incorrect description of our actual implementation (see Appendix A). We have rephrased this in the revision. For all data transformation methods, we carefully divided the training and test sets based on dates, ensuring that all transformations were applied exclusively to the training set. This guarantees no data leakage and ensures a fair comparison among all methods.

---

> ### Author Response · Authors · 2024-11-25
> **General Response (Part 2)**
>
> > 2. To Reviewer MUZC and jZAh: The differences between our conservative TD method and other ensemble-based Q-learning methods
>
> While the mathematical expressions of our method and other existing methods appear similar, significant differences exist in the network model and data input. Notably, existing ensemble-based Q-learning methods [1-3] typically utilize multiple target Q-networks (with separate model parameters) and compute ensemble value regularization by exploiting the **implicit diversity among these Q-networks**. In contrast, our approach relies on a single target Q-network and derives the worst-case Q-value by leveraging the **explicit diversity introduced through transformed data**.
>
> Specifically:
> - Existing methods are based on **model diversity**: Most existing ensemble-based methods require multiple Q-networks with identical input data $(s_{t+1}, a_{t+1})$ to calculate a conservative TD target, as shown in Eq. (1) below, which significantly increases the model size. For example,
>     - [1] uses the minimum value of multiple parallel Q-networks as the Bellman target;
>     - [2] stabilizes Q-learning by averaging previously learned Q-values as the target;
>     - [3] averages all Q-values, excluding those with the highest $N-K$ values.
> - Our method is based on **data diversity**: Our ensemble method is based on original stock data and its transformations $(s_{t+1}^{(n)}, a_{t+1}^{(n)})$ to calculate a conservative TD target by a single Q-function, as illustrated in Eq. (2) below. This approach leverages transformations of stock data to account for diverse market conditions, thereby capturing more variability in the decision-making process. We illustrate the feasibility of our approach from the perspective of Bellman Equations within our offline RL setup (Eq. 4 in the manuscript).
>
>
> $ \hat{Q}^\prime (s\_t, a\_t) = r\_t +\gamma\big[
>     -\lambda \log \pi\_{\theta} ({\hat{a}}\_{t+1} \mid {s}\_{t+1}) + \underset {k=1,\dots, M}{\Psi} {Q\_{\bar{\phi}\_k}({s}\_{t+1}, {\hat{a}}\_{t+1})}\big]$ (1)
>
> $ \hat{Q}^\prime (s_t, a_t) = r_t +\gamma\big[
>     -\lambda \log \pi_{\theta} ({\hat{a}}\_{t+1} \mid {s}\_{t+1}) + \underset {k=1,2}{\min} \ \underset {n=1:N}{\min} \big({Q\_{\bar{\phi}_k}({s}\_{t+1}, {\hat{a}}\_{t+1})}, Q\_{\bar{\phi}\_k}({s}\_{t+1}^{(n)}, {\hat{a}}\_{t+1}^{(n)})\big) \big]$  (2)
>
> References:
>
> [1] Uncertainty-Based Offline Reinforcement Learning with Diversified Q-Ensemble, NeurIPS 2021.
>
> [2] Offline-to-Online Reinforcement Learning via Balanced Replay and Pessimistic Q-Ensemble, CoRL 2021.
>
> [3] Aggressive Q-Learning with Ensembles: Achieving Both High Sample Efficiency and High Asymptotic Performance, NeurIPS 2022.
>
> > 3. To Reviewer hygz, jZAh: Experiments on a larger market scale
>
> **(1) Why is using a larger market scale challenging?**
>
> In this work, we focus on advancing RL-based stock trading. To the best of our knowledge, existing RL-based stock trading methods, such as FinRL, StockFormer, and SARL, primarily conduct experiments on relatively small-scale datasets. We attribute this limitation to two main factors:
> - Data perspective: Trading suspensions frequently occur in real-world stock data. Previous studies often select stocks based on the requirement that the proportion of valid data exceeds a specific threshold (e.g., 98% in StockFormer) to reduce noise from excessive data interpolation.
> - Algorithm perspective: As the stock pool size increases, the action space grows significantly, making it more challenging for RL methods to manage. If we aim to trade thousands of stocks in the market, the dimensionality of the action space can be even larger than the number of training sequences! The difficulty of high-dimensional action space is well-documented in other domains beyond stock trading [4,5]. However, learning an RL-for-stock model with a filtered small stock pool remains valuable. In practice, we can often manually pre-select a set of promising stocks using prior knowledge, enabling more focused and effective RL training.
>
> **(2) Additional results on an expanded dataset with 587 stocks**
>
> Nevertheless, we conduct experiments on a larger dataset by expanding the range of CSI stocks and selecting a dataset containing 587 stocks. We maintain the same experimental setup as in the offline evaluation described in the manuscript and compare our method with several baselines. The results are presented in the table below.
>
> | | Cumulative return (CR) $\uparrow$ | Annualized return (AR)  $\uparrow$ | Sharpe ratio (SR) $\uparrow$ | Maximum drawdown (MDD) $\downarrow$ |
> | - | - | - | - | - |
> | SARL  | 0.16| 0.05  | 0.28| 0.47 |
> | FinRL-SAC   | -0.12| -0.04 | -0.03 | 0.44|
> | StockFormer | 0.18| 0.06 | 0.32| 0.39|
> | MetaTrader  | **0.41** | **0.12** | **0.60** | **0.37**  |
>
>
> References:
>
> [4] Action Branching Architectures for Deep Reinforcement Learning, AAAI 2018.
>
> [5] POTEC: Off-Policy Learning for Large Action Spaces via Two-Stage Policy Decomposition, ICML 2023.

---

### Meta-Review · Area_Chair_Ts1R · 2024-12-20

**Metareview:**

The paper presents a reinforcement learning (RL) approach for stock trading. It uses data transformation techniques and a conservative TD ensemble to improve generalization in non-stationary markets. The methodology is well-structured, and the experiments show promising results, with improvements over existing models.

However, there are key concerns. The rationale behind the chosen data transformations, such as focusing only on the top 10% of stocks, is unclear and may limit the model's adaptability. Additionally, using small observation windows for covariance matrix estimation may introduce bias, and an ablation study to assess this would strengthen the paper. The comparison with ensemble Q-learning methods is also missing, which would provide further context for the approach.

In conclusion, the paper offers valuable insights but requires further clarification and experiments to address these issues.

**Additional Comments On Reviewer Discussion:**

After the rebuttal, reviewers remained divided. Some appreciated the authors' clarifications and additional experiments, while others still had concerns, particularly about the rationale behind data transformations, the potential bias in covariance matrix estimation, and the lack of comparison with relevant methods like ensemble Q-learning. These unresolved issues led some reviewers to maintain their original evaluation, while others remained supportive of the approach.
The decision to reject the paper is primarily based on unresolved concerns regarding the methodology and experimental setup.

---

### Decision · Program_Chairs · 2025-01-22

Reject